# Efficient Aggregated Kernel Tests using Incomplete $U$-statistics

**Antonin Schrab**
Centre for Artificial Intelligence
Gatsby Computational Neuroscience Unit
University College London & Inria London
`a.schrab@ucl.ac.uk`

**Ilmun Kim**
Department of Statistics & Data Science
Department of Applied Statistics
Yonsei University
`ilmun@yonsei.ac.kr`

**Benjamin Guedj**
Centre for Artificial Intelligence
University College London & Inria London
`b.guedj@ucl.ac.uk`

**Arthur Gretton**
Gatsby Computational Neuroscience Unit
University College London
`arthur.gretton@gmail.com`

## Abstract

We propose a series of computationally efficient nonparametric tests for the two-sample, independence, and goodness-of-fit problems, using the Maximum Mean Discrepancy (MMD), Hilbert Schmidt Independence Criterion (HSIC), and Kernel Stein Discrepancy (KSD), respectively. Our test statistics are incomplete $U$-statistics, with a computational cost that interpolates between linear time in the number of samples, and quadratic time, as associated with classical $U$-statistic tests. The three proposed tests aggregate over several kernel bandwidths to detect departures from the null on various scales: we call the resulting tests MMDAggInc, HSICAggInc and KSDAggInc. This procedure provides a solution to the fundamental kernel selection problem as we can aggregate a large number of kernels with several bandwidths without incurring a significant loss of test power. For the test thresholds, we derive a quantile bound for wild bootstrapped incomplete $U$-statistics, which is of independent interest. We derive non-asymptotic uniform separation rates for MMDAggInc and HSICAggInc, and quantify exactly the trade-off between computational efficiency and the attainable rates: this result is novel for tests based on incomplete $U$-statistics, to our knowledge. We further show that in the quadratic-time case, the wild bootstrap incurs no penalty to test power over the more widespread permutation-based approach, since both attain the same minimax optimal rates (which in turn match the rates that use oracle quantiles). We support our claims with numerical experiments on the trade-off between computational efficiency and test power. In all three testing frameworks, the linear-time versions of our proposed tests perform at least as well as the current linear-time state-of-the-art tests.

## 1   Introduction

Nonparametric hypothesis testing is a fundamental field of statistics, and is widely used by the machine learning community and practitioners in numerous other fields, due to the increasing availability of huge amounts of data. When dealing with large-scale datasets, computational cost can quickly emerge as a major issue which might prevent from using expensive tests in practice; constructing efficient tests is therefore crucial for their real-world applications. In this paper, we construct kernel-based aggregated tests using incomplete $U$-statistics (Blom, 1976) for the **two-sample**, **independence** and

36th Conference on Neural Information Processing Systems (NeurIPS 2022).

**goodness-of-fit** problems (which we detail in Section 2). The quadratic-time aggregation procedure has been shown to result in powerful tests (Fromont et al., 2012; Fromont et al., 2013; Albert et al., 2022; Schrab et al., 2021, 2022), we propose efficient variants of these well-studied tests, with computational cost interpolating from the classical quadratic-time regime to the linear-time one.

**Related work: aggregated tests.** Kernel selection (or kernel bandwidth selection) is a fundamental problem in nonparametric hypothesis testing as this choice has a major influence on test power. Motivated by this problem, non-asymptotic aggregated tests, which combine tests with different kernel bandwidths, have been proposed for the two-sample (Fromont et al., 2012, 2013; Kim et al., 2022; Schrab et al., 2021), independence (Albert et al., 2022; Kim et al., 2022), and goodness-of-fit (Schrab et al., 2022) testing frameworks. Li and Yuan (2019) and Balasubramanian et al. (2021) construct similar aggregated tests for these three problems, with the difference that they work in the asymptotic regime. All the mentioned works study aggregated tests in terms of uniform separation rates (Baraud, 2002). Those rates depend on the sample size and satisfy the following property: if the $L^2$-norm difference between the densities is greater than the uniform separation rate, then the test is guaranteed to have high power. All aggregated kernel-based tests in the existing literature have been studied using $U$-statistic estimators (Hoeffding, 1992) with tests running in quadratic time.

**Related work: efficient kernel tests.** Several linear-time kernel tests have been proposed for those three testing frameworks. Those include tests using classical linear-time estimators with median bandwidth (Gretton et al., 2012a; Liu et al., 2016) or selecting an optimal bandwidth on held-out data to maximize power (Gretton et al., 2012b), tests using eigenspectrum approximation (Gretton et al., 2009), tests using post-selection inference for adaptive kernel selection with incomplete $U$-statistics (Yamada et al., 2018, 2019; Lim et al., 2019, 2020; Kübler et al., 2020; Freidling et al., 2021), tests which use a Nyström approximation of the asymptotic null distribution (Zhang et al., 2018; Cherfaoui et al., 2022), random Fourier features tests (Zhang et al., 2018; Zhao and Meng, 2015; Chwialkowski et al., 2015), tests based on random feature Stein discrepancies (Huggins and Mackey, 2018), the adaptive tests which use features selected on held-out data to maximize power (Jitkrittum et al., 2016, 2017a,b), as well as tests using neural networks to learn a discrepancy (Grathwohl et al., 2020). We also point out the very relevant works of Kübler et al. (2022) on a quadratic-time test, and of Ho and Shieh (2006), Zaremba et al. (2013) and Zhang et al. (2018) on the use of block $U$-statistics with complexity $\mathcal{O}(N^{1.5})$ for block size $\sqrt{N}$ where $N$ is the sample size.

**Contributions and outline.** In Section 2, we present the three testing problems with their associated well-known quadratic-time kernel-based estimators (MMD, HSIC, KSD) which are $U$-statistics. We introduce three associated incomplete $U$-statistics estimators, which can be computed efficiently, in Section 3. We then provide quantile and variance bounds for generic incomplete $U$-statistics using a wild bootstrap, in Section 4. We study the level and power guarantees at every finite sample sizes for our efficient tests using incomplete $U$-statistics for a fixed kernel bandwidth, in Section 5. In particular, we obtain non-asymptotic uniform separation rates for the two-sample and independence tests over a Sobolev ball, and show that these rates are minimax optimal up to the cost incurred for efficiency of the test. In Section 6, we propose our efficient aggregated tests which combine tests with multiple kernel bandwidths. We prove that the proposed tests are adaptive over Sobolev balls and achieve the same uniform separation rate (up to an iterated logarithmic term) as the tests with optimal bandwidths. As a result of our analysis, we have shown minimax optimality over Sobolev balls of the quadratic-time tests using quantiles estimated with a wild bootstrap. Whether this optimality result also holds for tests using the more general permutation-based procedure to approximate HSIC quantiles, was an open problem formulated by Kim et al. (2022), we prove that it indeed holds in Section 7. As observed in Section 8, the linear-time versions of MMDAggInc, HSICAggInc and KSDAggInc retain high power, and either outperform or match the power of other state-of-the-art linear-time kernel tests. Our implementation of the tests and code for reproducibility of the experiments are available online under the MIT license: `https://github.com/antoninschrab/agginc-paper`.

## 2  Background

In this section, we briefly describe our main problems of interest, comprising the two-sample, independence and goodness-of-fit problems. We approach these problems from a nonparametric point of view using the kernel-based statistics: MMD, HSIC, and KSD. We briefly introduce original forms of these statistics, which can be computed in quadratic time, and also discuss ways of calibrating tests proposed in the literature. The three quadratic-time expressions are presented in Appendix B.

**Two-sample testing.** In this problem, we are given independent samples $\mathbb{X}_m := (X_i)_{1 \leq i \leq m}$ and $\mathbb{Y}_n = (Y_j)_{1 \leq j \leq n}$, consisting of i.i.d. random variables with respective probability density functions[1] $p$ and $q$ on $\mathbb{R}^d$. We assume we work with balanced sample sizes, that is[2] $\max(m,n) \lesssim \min(m,n)$. We are interested in testing the null hypothesis $\mathcal{H}_0 : p = q$ against the alternative $\mathcal{H}_1 : p \neq q$; that is, we want to know if the samples come from the same distribution. Gretton et al. (2012a) propose a nonparametric kernel test based on the *Maximum Mean Discrepancy* (MMD), a measure between probability distributions which uses a characteristic kernel $k$ (Fukumizu et al., 2008; Sriperumbudur et al., 2011). It can be estimated using a quadratic-time estimator (Gretton et al., 2012a, Lemma 6) which, as noted by Kim et al. (2022), can be expressed as a two-sample $U$-statistic (both of second order) (Hoeffding, 1992),

$$\widehat{\mathrm{MMD}}_k^2(\mathbb{X}_m, \mathbb{Y}_n) = \frac{1}{\left|\mathbf{i}_2^m\right|\left|\mathbf{i}_2^n\right|} \sum_{(i,i') \in \mathbf{i}_2^m} \sum_{(j,j') \in \mathbf{i}_2^n} h_k^{\mathrm{MMD}}(X_i, X_{i'}; Y_j, Y_{j'}), \tag{1}$$

where $\mathbf{i}_a^b$ with $a \leq b$ denotes the set of all $a$-tuples drawn without replacement from $\{1, \ldots, b\}$ so that $\left|\mathbf{i}_a^b\right| = b \cdots (b - a + 1)$, and where, for $x_1, x_2, y_1, y_2 \in \mathbb{R}^d$, we let

$$h_k^{\mathrm{MMD}}(x_1, x_2; y_1, y_2) := k(x_1, x_2) - k(x_1, y_2) - k(x_2, y_1) + k(y_1, y_2). \tag{2}$$

**Independence testing.** In this problem, we have access to i.i.d. pairs of samples $\mathbb{Z}_N := \left(Z_i\right)_{1 \leq i \leq N} = \left((X_i, Y_i)\right)_{1 \leq i \leq N}$ with joint probability density $p_{xy}$ on $\mathbb{R}^{d_x} \times \mathbb{R}^{d_y}$ and marginals $p_x$ on $\mathbb{R}^{d_x}$ and $p_y$ on $\mathbb{R}^{d_y}$. We are interested in testing $\mathcal{H}_0 : p_{xy} = p_x \otimes p_y$ against $\mathcal{H}_1 : p_{xy} \neq p_x \otimes p_y$; that is, we want to know if two components of the pairs of samples are independent or dependent. Gretton et al. (2005, 2008) propose a nonparametric kernel test based on the *Hilbert Schmidt Independence Criterion* (HSIC). It can be estimated using the quadratic-time estimator proposed by Song et al. (2012, Equation 5) which is a fourth-order one-sample $U$-statistic

$$\widehat{\mathrm{HSIC}}_{k,\ell}(\mathbb{Z}_N) = \frac{1}{\left|\mathbf{i}_4^N\right|} \sum_{(i,j,r,s) \in \mathbf{i}_4^N} h_{k,\ell}^{\mathrm{HSIC}}(Z_i, Z_j, Z_r, Z_s) \tag{3}$$

for characteristic kernels $k$ on $\mathbb{R}^{d_x}$ and $\ell$ on $\mathbb{R}^{d_y}$ (Gretton, 2015), and where for $z_a = (x_a, y_a) \in \mathbb{R}^{d_x} \times \mathbb{R}^{d_y}$, $a = 1, \ldots, 4$, we let

$$h_{k,\ell}^{\mathrm{HSIC}}(z_1, z_2, z_3, z_4) := \frac{1}{4} h_k^{\mathrm{MMD}}(x_1, x_2; x_3, x_4) h_\ell^{\mathrm{MMD}}(y_1, y_2; y_3, y_4). \tag{4}$$

**Goodness-of-fit testing.** For this problem, we are given a model density $p$ on $\mathbb{R}^d$ and i.i.d. samples $\mathbb{Z}_N := (Z_i)_{1 \leq i \leq N}$ drawn from a density $q$ on $\mathbb{R}^d$. The aim is again to test $\mathcal{H}_0 : p = q$ against $\mathcal{H}_1 : p \neq q$; that is, we want to know if the samples have been drawn from the model. Chwialkowski et al. (2016) and Liu et al. (2016) both construct a nonparametric goodness-of-fit test using the *Kernel Stein Discrepancy* (KSD). A quadratic-time KSD estimator can be computed as the second-order one-sample $U$-statistic,

$$\widehat{\mathrm{KSD}}_{p,k}^2(\mathbb{Z}_N) := \frac{1}{\left|\mathbf{i}_2^N\right|} \sum_{(i,j) \in \mathbf{i}_2^N} h_{k,p}^{\mathrm{KSD}}(Z_i, Z_j), \tag{5}$$

where the *Stein kernel* $h_{k,p}^{\mathrm{KSD}} : \mathbb{R}^d \times \mathbb{R}^d \to \mathbb{R}$ is defined as

$$\begin{aligned} h_{k,p}^{\mathrm{KSD}}(x, y) := &\left(\nabla \log p(x)^\top \nabla \log p(y)\right) k(x,y) + \nabla \log p(y)^\top \nabla_x k(x,y) \\ &+ \nabla \log p(x)^\top \nabla_y k(x,y) + \sum_{i=1}^d \frac{\partial}{\partial x_i \partial y_i} k(x,y). \end{aligned} \tag{6}$$

In order to guarantee consistency of the Stein goodness-of-fit test (Chwialkowski et al., 2016, Theorem 2.2), we assume that the kernel $k$ is $C_0$-universal (Carmeli et al., 2010, Definition 4.1) and that

$$\mathbb{E}_q\left[h_{k,p}^{\mathrm{KSD}}(z, z)\right] < \infty \qquad \text{and} \qquad \mathbb{E}_q\left[\left\|\nabla \log\left(\frac{p(z)}{q(z)}\right)\right\|_2^2\right] < \infty. \tag{7}$$

---

[1] All probability density functions in this paper are with respect to the Lebesgue measure.

[2] We use the notation $a \lesssim b$ when there exists a constant $C > 0$ such that $a \leq Cb$. We similarly use the notation $\gtrsim$. We write $a \asymp b$ if $a \lesssim b$ and $a \gtrsim b$. We also use the convention that all constants are generically denoted by $C$, even though they might be different.

**Quantile estimation.** Multiple strategies have been proposed to estimate the quantiles of test statistics under the null for these three tests. We primarily focus on the wild bootstrap approach (Chwialkowski et al., 2014), though our results also hold using a parametric bootstrap for the goodness-of-fit setting (Schrab et al., 2022). In Section 7, we show that the same uniform separation rates can be derived for HSIC quadratic-time tests using permutations instead of a wild bootstrap.

More details on MMD, HSIC, KSD, and on quantile estimation are provided in Appendix B.

## 3 Incomplete $U$-statistics for MMD, HSIC and KSD

As presented above, the quadratic-time statistics for the two-sample (MMD), independence (HSIC) and goodness-of-fit (KSD) problems can be rewritten as $U$-statistics with kernels $h_k^{\mathrm{MMD}}$, $h_{k,\ell}^{\mathrm{HSIC}}$ and $h_{k,p}^{\mathrm{KSD}}$, respectively. The computational cost of tests based on these $U$-statistics grows quadratically with the sample size. When working with very large sample sizes, as it is often the case in real-world uses of those tests, this quadratic cost can become very problematic, and faster alternative tests are better adapted to this 'big data' setting. Multiple linear-time kernel tests have been proposed in the three testing frameworks (see Section 1 for details). We construct computationally efficient variants of the aggregated kernel tests proposed by Fromont et al. (2013), Albert et al. (2022), Kim et al. (2022), and Schrab et al. (2021, 2022) for the three settings, with the aim of retaining the significant power advantages of the aggregation procedure observed for quadratic-time tests. To this end, we propose to replace the quadratic-time $U$-statistics presented in Equations (1), (3) and (5) with second-order incomplete $U$-statistics (Blom, 1976; Janson, 1984; Lee, 1990),

$$\overline{\mathrm{MMD}}_k^2\big(\mathbb{X}_m, \mathbb{Y}_n; \mathcal{D}_N\big) \coloneqq \frac{1}{\big|\mathcal{D}_N\big|} \sum_{(i,j)\in\mathcal{D}_N} h_k^{\mathrm{MMD}}(X_i, X_j; Y_i, Y_j), \tag{8}$$

$$\overline{\mathrm{HSIC}}_{k,\ell}\big(\mathbb{Z}_N; \mathcal{D}_{\lfloor N/2 \rfloor}\big) \coloneqq \frac{1}{\big|\mathcal{D}_{\lfloor N/2 \rfloor}\big|} \sum_{(i,j)\in\mathcal{D}_{\lfloor N/2 \rfloor}} h_{k,\ell}^{\mathrm{HSIC}}\big(Z_i, Z_j, Z_{i+\lfloor N/2 \rfloor}, Z_{j+\lfloor N/2 \rfloor}\big), \tag{9}$$

$$\overline{\mathrm{KSD}}_{p,k}^2\big(\mathbb{Z}_N; \mathcal{D}_N\big) \coloneqq \frac{1}{\big|\mathcal{D}_N\big|} \sum_{(i,j)\in\mathcal{D}_N} h_{k,p}^{\mathrm{KSD}}(Z_i, Z_j), \tag{10}$$

where for the two-sample problem we let $N \coloneqq \min(m, n)$, and where the *design* $\mathcal{D}_b$ is a subset of $\mathbf{i}_2^b$ (the set of all 2-tuples drawn without replacement from $\{1, \ldots, b\}$). Note that $\mathcal{D}_{\lfloor N/2 \rfloor} \subseteq \mathbf{i}_2^{N/2} \subset \mathbf{i}_2^N$. The design can be deterministic. For example, for the two-sample problem with equal even sample sizes $m = n = N$, the deterministic design $\mathcal{D}_N = \{(2a - 1, 2a) : a = 1, \ldots, N/2\}$ corresponds to the MMD linear-time estimator proposed by Gretton et al. (2012a, Lemma 14). For fixed design size, the elements of the design can also be chosen at random without replacement, in which case the estimators in Equations (8) to (10) become random quantities given the data. For generality purposes, the results presented in this paper hold for both deterministic and random (without replacement) design choices while we focus on the deterministic design in our experiments. By fixing the design sizes in Equations (8) to (10) to be, for example,

$$\big|\mathcal{D}_N\big| = \big|\mathcal{D}_{\lfloor N/2 \rfloor}\big| = cN \tag{11}$$

for some small constant $c \in \mathbb{N} \setminus \{0\}$, we obtain incomplete $U$-statistics which can be computed in linear time. Note that by pairing the samples $Z_i \coloneqq (X_i, Y_i)$, $i = 1, \ldots, N$ for the MMD case and $\widetilde{Z}_i \coloneqq \big(Z_i, Z_{i+\lfloor N/2 \rfloor}\big)$, $i = 1, \ldots, \lfloor N/2 \rfloor$ for the HSIC case, we observe that all three incomplete $U$-statistics of second order have the same form, with only the kernel functions and the design differing. The motivation for defining the estimators in Equations (8) and (9) as incomplete $U$-statistics of order 2 (rather than of higher order) derives from the reasoning of Kim et al. (2022, Section 6) for permuted complete $U$-statistics for the two-sample and independence problems (see Appendix E.1).

## 4 Quantile and variance bounds for incomplete $U$-statistics

In this section, we derive upper quantile and variance bounds for a second-order incomplete degenerate $U$-statistic with a generic degenerate kernel $h$, for some design $\mathcal{D} \subseteq \mathbf{i}_2^N$, defined as

$$\overline{U}\big(\mathbb{Z}_N; \mathcal{D}\big) \coloneqq \frac{1}{|\mathcal{D}|} \sum_{(i,j)\in\mathcal{D}} h(Z_i, Z_j).$$

We will use these results to bound the quantiles and variances of our three test statistics for our hypothesis tests in Section 5. The derived bounds are of independent interest.

In the following lemma, building on the results of Lee (1990), we directly derive an upper bound on the variance of the incomplete $U$-statistic in terms of the sample size $N$ and of the design size $|\mathcal{D}|$.

**Lemma 1.** *The variance of the incomplete $U$-statistic can be upper bounded in terms of the quantities $\sigma_1^2 := \mathrm{var}\big(\mathbb{E}\big[h(Z, Z')\big| Z'\big]\big)$ and $\sigma_2^2 := \mathrm{var}(h(Z, Z'))$ with different bounds depending on the design choice. For deterministic (LHS) or random (RHS) design $\mathcal{D}$ and sample size $N$, we have*

$$
\mathrm{var}\big(\overline{U}\big) \lesssim \frac{N}{|\mathcal{D}|}\sigma_1^2 + \frac{1}{|\mathcal{D}|}\sigma_2^2 \qquad and \qquad \mathrm{var}\big(\overline{U}\big) \lesssim \frac{1}{N}\sigma_1^2 + \frac{1}{|\mathcal{D}|}\sigma_2^2.
$$

The proof of Lemma 1 is deferred to Appendix F.2. We emphasize the fact that this variance bound also holds for random design with replacement, as considered by Blom (1976) and Lee (1990). For random design, we observe that if $|\mathcal{D}| \asymp N^2$ then the bound is $\sigma_1^2/N + \sigma_2^2/N^2$ which is the variance bound of the complete $U$-statistic (Albert et al., 2022, Lemma 10). If $N \lesssim |\mathcal{D}| \lesssim N^2$, the variance bound is $\sigma_1^2/N + \sigma_2^2/|\mathcal{D}|$, and if $|\mathcal{D}| \lesssim N$ it is $\sigma_2^2/|\mathcal{D}|$ since $\sigma_1^2 \leq \sigma_2^2/2$ (Blom, 1976, Equation 2.1).

Kim et al. (2022) develop exponential concentration bounds for permuted complete $U$-statistics, and Clémençon et al. (2013) study the uniform approximation of $U$-statistics by incomplete $U$-statistics. To the best of our knowledge, no quantile bounds have yet been obtained for incomplete $U$-statistics in the literature. While permutations are well-suited for complete $U$-statistics (Kim et al., 2022), using them with incomplete $U$-statistics results in having to compute new kernel values, which comes at an additional computational cost we would like to avoid. Restricting the set of permutations to those for which the kernel values have already been computed for the original incomplete $U$-statistic corresponds exactly to using a wild bootstrap (Schrab et al., 2021, Appendix B). Hence, we consider the wild bootstrapped second-order incomplete $U$-statistic

$$
\overline{U}^\epsilon\big(\mathbb{Z}_N; \mathcal{D}\big) := \frac{1}{|\mathcal{D}|} \sum_{(i,j)\in\mathcal{D}} \epsilon_i \epsilon_j h(Z_i, Z_j) \tag{12}
$$

for i.i.d. Rademacher random variables $\epsilon_1, \ldots, \epsilon_N$ with values in $\{-1, 1\}$, for which we derive an exponential concentration bound (quantile bound). We note the in-depth work of Chwialkowski et al. (2014) on the wild bootstrap procedure for kernel tests with applications to quadratic-time MMD and HSIC tests. We now provide exponential tail bounds for wild bootstrapped incomplete $U$-statistics.

**Lemma 2.** *There exists some constant $C > 0$ such that, for every $t \geq 0$, we have*

$$
\mathbb{P}_\epsilon\Big(\big|\overline{U}^\epsilon\big| \geq t \,\big|\, \mathbb{Z}_N, \mathcal{D}\Big) \ \leq\ 2\exp\Big(-C\frac{t}{A_{\mathrm{inc}}}\Big) \ \leq\ 2\exp\Big(-C\frac{t}{A}\Big)
$$

*where $A_{\mathrm{inc}}^2 := |\mathcal{D}|^{-2} \sum_{(i,j)\in\mathcal{D}} h(Z_i, Z_j)^2$ and $A^2 := |\mathcal{D}|^{-2} \sum_{(i,j)\in\mathbf{i}_2^N} h(Z_i, Z_j)^2$.*

Lemma 2 is proved in Appendix F.3. While the second bound in Lemma 2 is less tight, it has the benefit of not depending on the choice of design $\mathcal{D}$ but only on its size $|\mathcal{D}|$ which is usually fixed.

## 5 Efficient kernel tests using incomplete $U$-statistics

We now formally define the hypothesis tests obtained using the incomplete $U$-statistics with a wild bootstrap. This is done for fixed kernel bandwidths $\lambda \in (0, \infty)^{d_x}, \mu \in (0, \infty)^{d_y}$, for the kernels[3]

$$
k_\lambda(x, y) := \prod_{i=1}^{d_x} \frac{1}{\lambda_i} K_i\left(\frac{x_i - y_i}{\lambda_i}\right), \qquad \ell_\mu(x, y) := \prod_{i=1}^{d_y} \frac{1}{\mu_i} L_i\left(\frac{x_i - y_i}{\mu_i}\right), \tag{13}
$$

for characteristic kernels $(x, y) \mapsto K_i(x - y), (x, y) \mapsto L_i(x - y)$ on $\mathbb{R} \times \mathbb{R}$ for functions $K_i, L_i \in L^1(\mathbb{R}) \cap L^2(\mathbb{R})$ integrating to 1. We unify the notation for the three testing frameworks. For the two-sample and goodness-of-fit problems, we work only with $k_\lambda$ and have $d = d_x$. For the independence

---

[3]Our results are presented for bandwidth selection, but they hold in the more general setting of kernel selection, as considered by Schrab et al. (2022). The goodness-of-fit results hold for a wider range of kernels including the IMQ (inverse multiquadric) kernel (Gorham and Mackey, 2017), as in Schrab et al. (2022).

problem, we work with the two kernels $k_\lambda$ and $\ell_\mu$, and for ease of notation we let $d := d_x + d_y$ and $\lambda_{d_x+i} := \mu_i$ for $i = 1, \dots, d_y$. We also simply write $p := p_{xy}$ and $q := p_x \otimes p_y$. We let $\overline{U}_\lambda$ and $h_\lambda$ denote either $\overline{\mathrm{MMD}}^2_{k_\lambda}$ and $h^{\mathrm{MMD}}_{k_\lambda}$, or $\overline{\mathrm{HSIC}}_{k_\lambda,\ell_\mu}$ and $h^{\mathrm{HSIC}}_{k_\lambda,\ell_\mu}$, or $\overline{\mathrm{KSD}}^2_{p,k_\lambda}$ and $h^{\mathrm{KSD}}_{k_\lambda,p}$, respectively. We denote the design size of the incomplete $U$-statistics in Equations (8) to (10) by

$$L := |\mathcal{D}_N| = |\mathcal{D}_{\lfloor N/2 \rfloor}|.$$

For the three testing frameworks, we estimate the quantiles of the test statistics by simulating the null hypothesis using a wild bootstrap, as done in the case of complete $U$-statistics by Fromont et al. (2012) and Schrab et al. (2021) for the two-sample problem, and by Schrab et al. (2022) for the goodness-of-fit problem. This is done by considering the original test statistic $U^{B_1+1}_\lambda := \overline{U}_\lambda$ together with $B_1$ wild bootstrapped incomplete $U$-statistics $U^1_\lambda, \dots, U^{B_1}_\lambda$ computed as in Equation (12), and estimating the $(1-\alpha)$-quantile with a Monte Carlo approximation

$$\widehat{q}^\lambda_{1-\alpha} := \inf\left\{t \in \mathbb{R} : 1 - \alpha \le \frac{1}{B_1 + 1} \sum_{b=1}^{B_1+1} \mathbf{1}\big(U^b_\lambda \le t\big)\right\} = U^{\bullet\lceil B_1(1-\alpha)\rceil}_\lambda, \qquad (14)$$

where $U^{\bullet 1}_\lambda \le \cdots \le U^{\bullet B_1+1}_\lambda$ are the sorted elements $U^1_\lambda, \dots, U^{B_1+1}_\lambda$. The test $\Delta^\lambda_\alpha$ is defined as rejecting the null if the original test statistic $\overline{U}_\lambda$ is greater than the estimated $(1-\alpha)$-quantile, that is,

$$\Delta^\lambda_\alpha(\mathbb{Z}_N) := \mathbf{1}\big(\overline{U}_\lambda(\mathbb{Z}_N) > \widehat{q}^\lambda_{1-\alpha}\big).$$

The resulting test has time complexity $\mathcal{O}(B_1 L)$ where $L$ is the design size ($1 \le L \le N(N-1)$). We show in Proposition 1 that the test $\Delta^\lambda_\alpha$ has well-calibrated asymptotic level for goodness-of-fit testing, and well-calibrated non-asymptotic level for two-sample and independence testing. The proof of the latter non-asymptotic guarantee is based on the exchangeability of $U^1_\lambda, \dots, U^{B_1+1}_\lambda$ under the null hypothesis along with the result of Romano and Wolf (2005, Lemma 1). A similar proof strategy can be found in Fromont et al. (2012, Proposition 2), Albert et al. (2022, Proposition 1), and Schrab et al. (2021, Proposition 1). The exchangeability of wild bootstrapped incomplete $U$-statistics for independence testing does not follow directly from the mentioned works. We show this through the interesting connection between $h^{\mathrm{HSIC}}_{k,\ell}$ and $\{h^{\mathrm{MMD}}_k, h^{\mathrm{MMD}}_\ell\}$, the proof is deferred to Appendix F.1.

**Proposition 1.** *The test $\Delta^\lambda_\alpha$ has level $\alpha \in (0,1)$, i.e. $\mathbb{P}_{\mathcal{H}_0}\big(\Delta^\lambda_\alpha(\mathbb{Z}_N) = 1\big) \le \alpha$. This holds non-asymptotically for the two-sample and independence cases, and asymptotically for goodness-of-fit.*[4]

Having established the validity of the test $\Delta^\lambda_\alpha$, we now study power guarantees for it in terms of the $L^2$-norm of the difference in densities $\|p - q\|_2$. In Theorem 1, we show for the three tests that, if $\|p - q\|_2$ exceeds some threshold, we can guarantee high test power. For the two-sample and independence problems, we derive uniform separation rates (Baraud, 2002) over Sobolev balls

$$\mathcal{S}^s_d(R) := \left\{f \in L^1(\mathbb{R}^d) \cap L^2(\mathbb{R}^d) : \int_{\mathbb{R}^d} \|\xi\|^{2s}_2 |\widehat{f}(\xi)|^2 \mathrm{d}\xi \le (2\pi)^d R^2\right\}, \qquad (15)$$

with radius $R > 0$ and smoothness parameter $s > 0$, where $\widehat{f}$ denotes the Fourier transform of $f$. The uniform separation rate over $\mathcal{S}^s_d(R)$ is the smallest value of $t$ such that, for any alternative with $\|p - q\|_2 > t$ and[5] $p - q \in \mathcal{S}^s_d(R)$, the probability of type II error of $\Delta^\lambda_\alpha$ can be controlled by $\beta \in (0,1)$. Before presenting Theorem 1, we introduce further notation unified over the three testing frameworks; we define the integral transform $T_\lambda$ as

$$(T_\lambda f)(x) := \int_{\mathbb{R}^d} f(x) \mathcal{K}_\lambda(x,y) \, \mathrm{d}y \qquad (16)$$

for $f \in L^2(\mathbb{R}^d)$, $x \in \mathbb{R}^d$, where $\mathcal{K}_\lambda := k_\lambda$ for the two-sample problem, $\mathcal{K}_\lambda := k_\lambda \otimes \ell_\mu$ for the independence problem, and $\mathcal{K}_\lambda := h^{\mathrm{KSD}}_{k_\lambda,p}$ for the goodness-of-fit problem. Note that, for the two-sample and independence testing frameworks, since $\mathcal{K}_\lambda$ is translation-invariant, the integral transform corresponds to a convolution. However, this is not true for the goodness-of-fit setting as $h^{\mathrm{KSD}}_{k_\lambda,p}$ is not translation-invariant. We are now in a position to present our main contribution in Theorem 1; we derive power guarantee conditions for our tests using incomplete $U$-statistics, and uniform separation rates over Sobolev balls for the two-sample and independence settings.

---

[4]Level is non-asymptotic for the goodness-of-fit case using a parametric bootstrap (Schrab et al., 2022). For the goodness-of-fit setting, we also recall that the further assumptions in Equation (7) need to be satisfied.

[5]We stress that we only assume $p - q \in \mathcal{S}^s_d(R)$ and not $p, q \in \mathcal{S}^s_d(R)$ as considered by Li and Yuan (2019). Viewing $q$ as a perturbed version of $p$, we only require that the perturbation is smooth (*i.e.* lies in a Sobolev ball).

**Theorem 1.** *Suppose that the assumptions in Appendix A.1 hold, and consider $\lambda \in (0, \infty)^d$.*

*(i) For sample size $N$ and design size $L$, if there exists some $C > 0$ such that*

$$\|p - q\|_2^2 \;\geq\; \|(p - q) - T_\lambda(p - q)\|_2^2 + C \frac{N}{L} \frac{\ln(1/\alpha)}{\beta} \sigma_{2,\lambda},$$

*then $\mathbb{P}_{\mathcal{H}_1}\big(\Delta_\alpha^\lambda(\mathbb{Z}_N) = 0\big) \leq \beta$ (type II error), where $\sigma_{2,\lambda} \lesssim 1/\sqrt{\lambda_1 \cdots \lambda_d}$ for MMD and HSIC.*

*(ii) Fix $R > 0$ and $s > 0$, and consider the bandwidths $\lambda_i^* := (N/L)^{2/(4s+d)}$ for $i = 1, \ldots, d$. For MMD and HSIC, the uniform separation rate of $\Delta_\alpha^{\lambda^*}$ over the Sobolev ball $\mathcal{S}_d^s(R)$ is (up to a constant)*

$$\big(L/N\big)^{-2s/(4s+d)}.$$

The proof of Theorem 1 relies on the variance and quantile bounds presented in Lemmas 1 and 2, and also uses results of Albert et al. (2022) and Schrab et al. (2021, 2022) on complete $U$-statistics. The details can be found in Appendix F.4. The power condition in Theorem 1 (i) corresponds to a variance-bias decomposition; for large bandwidths the bias term (first term) dominates, while for small bandwidths the variance term (second term which also controls the quantile) dominates. While the power guarantees of Theorem 1 hold for any design (either deterministic or uniformly random without replacement) of fixed size $L$, the choice of design still influences the performance of the test in practice. The variance (but not its upper bound) depends on the choice of design; certain choices lead to minimum variance of the incomplete $U$-statistic (Lee, 1990, Section 4.3.2).

The minimax (*i.e.* optimal) rate over the Sobolev ball $\mathcal{S}_d^s(R)$ is $N^{-2s/(4s+d)}$ for the two-sample (Li and Yuan, 2019, Theorem 5 (ii)) and independence (Albert et al., 2022, Theorem 4; Berrett et al., 2021, Corollary 5) problems. The rate for our incomplete $U$-statistic test with time complexity $\mathcal{O}(B_1 L)$ has the same dependence in the exponent as the minimax rate; $(L/N)^{-2s/(4s+d)} = N^{-2s/(4s+d)} \big(N^2/L\big)^{2s/(4s+d)}$ where $L \lesssim N^2$ with $L$ the design size and $N$ the sample size.

---

- If $L \asymp N^2$ then the test runs in quadratic time and we recover exactly the minimax rate.
- If $N \lesssim L \lesssim N^2$ then the rate still converges to 0; there is a trade-off between the cost $(N^2/L)^{2s/(4s+d)}$ incurred in the minimax rate and the computational efficiency $\mathcal{O}(B_1 L)$.
- If $L \lesssim N$ then there is no guarantee that the rate converges to 0.

---

To summarize, the tests we propose have computational cost $\mathcal{O}(B_1 L)$ which can be specified by the user with the choice of the number of wild bootstraps $B_1$, and of the design size $L$ (as a function of the sample size $N$). There is a trade-off between test power and computational cost. We provide theoretical rates in terms of $L$ and $N$, working up to a constant. The rate is minimax optimal in the case where $L$ grows quadratically with $N$. We quantify exactly how, as the computational cost decreases from quadratic to linear in the sample size, the rate deteriorates gradually from being minimax optimal to not being guaranteed to convergence to zero. In our experiments, we use a design size which grows linearly with the sample size in order to compare our tests against other linear-time tests in the literature. The assumption guaranteeing that the rate converges to 0 is not satisfied in this setting, however, it would be satisfied for any faster growth of the design size (*e.g.* $L \asymp N \log \log N$).

## 6 Efficient aggregated kernel tests using incomplete $U$-statistics

We now introduce our aggregated tests that combine single tests with different bandwidths. Our aggregation scheme is similar to those of Fromont et al. (2013), Albert et al. (2022) and Schrab et al. (2021, 2022), and can yield a test which is adaptive to the unknown smoothness parameter $s$ of the Sobolev ball $\mathcal{S}_d^s(R)$, with relatively low price. Let $\Lambda$ be a finite collection of bandwidths, $(w_\lambda)_{\lambda \in \Lambda}$ be associated weights satisfying $\sum_{\lambda \in \Lambda} w_\lambda \leq 1$, and $u_\alpha$ be some correction term defined shortly in Equation (17). Then, using the incomplete $U$-statistic $\overline{U}_\lambda$, we define our aggregated test $\Delta_\alpha^\Lambda$ as

$$\Delta_\alpha^\Lambda(\mathbb{Z}_N) := \mathbf{1}\Big(\overline{U}_\lambda(\mathbb{Z}_N) > \widehat{q}_{1-u_\alpha w_\lambda}^\lambda \text{ for some } \lambda \in \Lambda\Big).$$

The levels of the single tests are weighted and adjusted with a correction term

$$u_\alpha := \sup_{B_3}\left\{ u \in \Big(0, \min_{\lambda \in \Lambda} w_\lambda^{-1}\Big) : \frac{1}{B_2} \sum_{b=1}^{B_2} \mathbf{1}\Big( \max_{\lambda \in \Lambda} \big( \widetilde{U}_\lambda^b - U_\lambda^{\bullet \lceil B_1(1 - u w_\lambda) \rceil} \big) > 0 \Big) \leq \alpha \right\}, \quad (17)$$

where the wild bootstrapped incomplete $U$-statistics $\widetilde{U}_\lambda^1, \ldots, \widetilde{U}_\lambda^{B_2}$ computed as in Equation (12) are used to perform a Monte Carlo approximation of the probability under the null, and where the supremum is estimated using $B_3$ steps of bisection method. Proposition 1, along with the reasoning of Schrab et al. (2021, Proposition 8), ensures that $\Delta_\alpha^\Lambda$ has non-asymptotic level $\alpha$ for the two-sample and independence cases, and asymptotic level $\alpha$ for the goodness-of-fit case. We refer to the three aggregated tests constructed using incomplete $U$-statistics as MMDAggInc, HSICAggInc and KSDAggInc. The computational complexity of those tests is $\mathcal{O}(|\Lambda|(B_1 + B_2)L)$, which means, for example, that if $L \asymp N$ as in Equation (11), the tests run efficiently in linear time in the sample size.

We formally record error guarantees of $\Delta_\alpha^\Lambda$ and derive uniform separation rates over Sobolev balls.

**Theorem 2.** *Suppose that the assumptions in Appendix A.2 hold, and consider a collection $\Lambda$.*

*(i) For sample size $N$ and design size $L$, if there exists some $C > 0$ such that*

$$\|p - q\|_2^2 \geq \min_{\lambda \in \Lambda} \left( \|(p - q) - T_\lambda(p - q)\|_2^2 + C \frac{N}{L} \frac{\ln\big(1/(\alpha w_\lambda)\big)}{\beta} \sigma_{2,\lambda} \right),$$

*then $\mathbb{P}_{\mathcal{H}_1}\big(\Delta_\alpha^\Lambda(\mathbb{Z}_N) = 0\big) \leq \beta$ (type II error), where $\sigma_{2,\lambda} \lesssim 1/\sqrt{\lambda_1 \cdots \lambda_d}$ for MMD and HSIC.*

*(ii) Assume $L > N$ so that $\ln(\ln(L/N))$ is well-defined. Consider the collections of bandwidths and weights (independent of the parameters $s$ and $R$ of the Sobolev ball $\mathcal{S}_d^s(R)$)*

$$\Lambda := \left\{ (2^{-\ell}, \ldots, 2^{-\ell}) \in (0, \infty)^d : \ell \in \left\{ 1, \ldots, \left\lceil \frac{2}{d} \log_2\left( \frac{L/N}{\ln(\ln(L/N))} \right) \right\rceil \right\} \right\}, \quad w_\lambda := \frac{6}{\pi^2 \ell^2}.$$

*For the two-sample and independence problems, the uniform separation rate of $\Delta_\alpha^\Lambda$ over the Sobolev balls $\big\{\mathcal{S}_d^s(R) : R > 0, s > 0\big\}$ is (up to a constant)*

$$\left( \frac{L/N}{\ln(\ln(L/N))} \right)^{-2s/(4s+d)}.$$

The extension from Theorem 1 to Theorem 2 has been proved for complete $U$-statistics in the two-sample (Fromont et al., 2013; Schrab et al., 2021), independence (Albert et al., 2022) and goodness-of-fit (Schrab et al., 2022) testing frameworks. The proof of Theorem 2 follows with the same reasoning by simply replacing $N$ with $L/N$ as we work with incomplete $U$-statistics; this 'replacement' is theoretically justified by Theorem 1. Theorem 2 shows that the aggregated test $\Delta_\alpha^\Lambda$ is *adaptive* over Sobolev balls $\big\{\mathcal{S}_d^s(R) : R > 0, s > 0\big\}$: the test $\Delta_\alpha^\Lambda$ does not depend on the unknown smoothness parameter $s$ (unlike $\Delta_\alpha^{\lambda^*}$ in Theorem 1) and achieves the minimax rate, up to an iterated logarithmic factor, and up to the cost incurred for efficiency of the test (*i.e.* $L/N$ instead of $N$).

## 7 Minimax optimal permuted quadratic-time aggregated independence test

Considering Theorem 2 with our incomplete $U$-statistic with full design $\mathcal{D} = \mathbf{i}_2^N$ for which $L \asymp N^2$, we have proved that the quadratic-time two-sample and independence aggregated tests using a wild bootstrap achieve the rate $(N/\ln(\ln(N)))^{-2s/(4s+d)}$ over the Sobolev balls $\big\{\mathcal{S}_d^s(R) : R > 0, s > 0\big\}$. This is the minimax rate (Li and Yuan, 2019; Albert et al., 2022), up to some iterated logarithmic term. For the two-sample problem, Kim et al. (2022) and Schrab et al. (2021) show that this optimality result also holds when using complete $U$-statistics with permutations. Whether the equivalent statement for the independence test with permutations holds has not yet been addressed; the rate can be proved using theoretical (unknown) quantiles with a Gaussian kernel (Albert et al., 2022), but has not yet been proved using permutations. Kim et al. (2022, Proposition 8.7) consider this problem, again using a Gaussian kernel, but they do not obtain the correct dependence on $\alpha$ (*i.e.* they obtain $\alpha^{-1/2}$ rather than $\ln(1/\alpha)$), hence they cannot recover the desired rate. As pointed out by Kim et al. (2022, Section 8): 'It remains an open question as to whether [the power guarantee] continues to hold when $\alpha^{-1/2}$ is replaced by $\ln(1/\alpha)$'. We now prove that we can improve the $\alpha$-dependence to $\ln(1/\alpha)^{3/2}$ for any bounded kernel of the form of Equation (13), and that this allows us to obtain the desired rate over Sobolev balls $\big\{\mathcal{S}_d^s(R) : R > 0, s > d/4\big\}$. The assumption $s > d/4$ imposes a stronger smoothness restriction on $p - q \in \mathcal{S}_d^s(R)$, which is similarly also considered by Li and Yuan (2019).

**Theorem 3.** *Consider the quadratic-time independence test using the complete U-statistic HSIC estimator with a quantile estimated using permutations as done by* Kim et al. (2022, Proposition 8.7), *with kernels as in Equation* (13) *for bounded functions $K_i$ and $L_j$ for $i = 1, \ldots, d_x$, $j = 1, \ldots, d_y$.*

*(i) Suppose that the assumptions in Appendix A.1 hold. For fixed $R > 0$, $s > d/4$, and bandwidths $\lambda_i^* := N^{-2/(4s+d)}$ for $i = 1, \ldots, d$, the probability of type II error of the test is controlled by $\beta$ when*

$$\|p - q\|_2^2 \geq \|(p-q) - T_{\lambda^*}(p-q)\|_2^2 + C\frac{1}{N}\frac{\ln(1/\alpha)^{3/2}}{\beta\sqrt{\lambda_1^* \cdots \lambda_d^*}} \quad \text{for some constant } C > 0.$$

*The uniform separation rate over the Sobolev ball $\mathcal{S}_d^s(R)$ is, up to a constant, $N^{-2s/(4s+d)}$.*

*(ii) Suppose that the assumptions in Appendix A.2 hold. The uniform separation rate over the Sobolev balls $\{\mathcal{S}_d^s(R) : R > 0, s > d/4\}$ is $(N/\ln(\ln(N)))^{-2s/(4s+d)}$, up to a constant, with the collections*

$$\Lambda := \left\{(2^{-\ell}, \ldots, 2^{-\ell}) \in (0, \infty)^d : \ell \in \left\{1, \ldots, \left\lceil\frac{2}{d}\log_2\left(\frac{N}{\ln(\ln(N))}\right)\right\rceil\right\}\right\}, \quad w_\lambda := \frac{6}{\pi^2\ell^2}.$$

The proof of Theorem 3, in Appendix F.5, uses the exponential concentration bound of Kim et al. (2022, Theorem 6.3) for permuted complete U-statistics. Another possible approach to obtain the correct dependency on $\alpha$ is to employ the sample-splitting method proposed by Kim et al. (2022, Section 8.3) in order to transform the independence problem into a two-sample problem. While this indirect approach leads to a logarithmic factor in $\alpha$, the practical power would be suboptimal due to an inefficient use of the data from sample splitting. Theorem 3 (i) shows that a $\ln(1/\alpha)^{3/2}$ dependence is achieved by the more practical permutation-based HSIC test. Theorem 3 (ii) demonstrates that this leads to a minimax optimal rate for the aggregated HSIC test, up to the $\ln(\ln(N))$ cost for adaptivity.

# 8 Experiments

For the two-sample problem, we consider testing samples drawn from a uniform density on $[0,1]^d$ against samples drawn from a perturbed uniform density. For the independence problem, the joint density is a perturbed uniform density on $[0,1]^{d_x+d_y}$, the marginals are then simply uniform densities. Those perturbed uniform densities can be shown to lie in Sobolev balls (Li and Yuan, 2019; Albert et al., 2022), to which our tests are adaptive. For the goodness-of-fit problem, we use a Gaussian-Bernoulli Restricted Boltzmann Machine as first considered by Liu et al. (2016) in this testing framework. We use collections of 21 bandwidths for MMD and HSIC and of 25 bandwidth pairs for HSIC; more details on the experiments (*e.g.* model and test parameters) are presented in Appendix C.

We consider our incomplete aggregated tests MMDAggInc, HSICAggInc and KSDAggInc, with parameter $R \in \{1, \ldots, N-1\}$ which fixes the deterministic design to consist of the first $R$ sub-diagonals of the $N \times N$ matrix, *i.e.* $\mathcal{D} := \{(i, i+r) : i = 1, \ldots, N-r \text{ for } r = 1, \ldots, R\}$ with size $|\mathcal{D}| = RN - R(R-1)/2$. We run our incomplete tests with $R \in \{1, 100, 200\}$ and also the complete test using the full design $\mathcal{D} = \mathbf{i}_2^N$. We compare their performances with current linear-time state-of-the-art tests: ME, SCF, FSIC and FSSD (Jitkrittum et al., 2016, 2017a,b) which evaluate the witness functions at a finite set of locations chosen to maximize the power, Cauchy RFF (random Fourier feature) and L1 IMQ (Huggins and Mackey, 2018) which are random feature Stein discrepancies, LSD (Grathwohl et al., 2020) which uses a neural network to learn the Stein discrepancy, and OST PSI (Kübler et al., 2020) which performs kernel selection using post selection inference.

Similar trends are observed across all our experiments in Figure 1, for the three testing frameworks, when varying the sample size, the dimension, and the difficulty of the problem (scale of perturbations or noise level). The linear-time tests AggInc $R = 200$ almost match the power obtained by the quadratic-time tests AggCom in all settings (except in Figure 1 (i) where the difference is larger) while being computationally much more efficient as can be seen in Figure 1 (d, h, l). The incomplete tests with $R = 100$ have power only slightly below the ones using $R = 200$, and run roughly twice as fast (Figure 1 (d, h, l)). In all experiments, those three tests (AggInc $R = 100, 200$ and AggCom) have significantly higher power than the linear-time tests which optimize test locations (ME, SCF, FSIC and FSSD); in the two-sample case the aggregated tests run faster for small sample size but slower for large sample size, in the independence case the aggregated tests run much faster, and in the goodness-of-fit case FSSD runs faster. While both types of tests are linear, we note that the runtimes

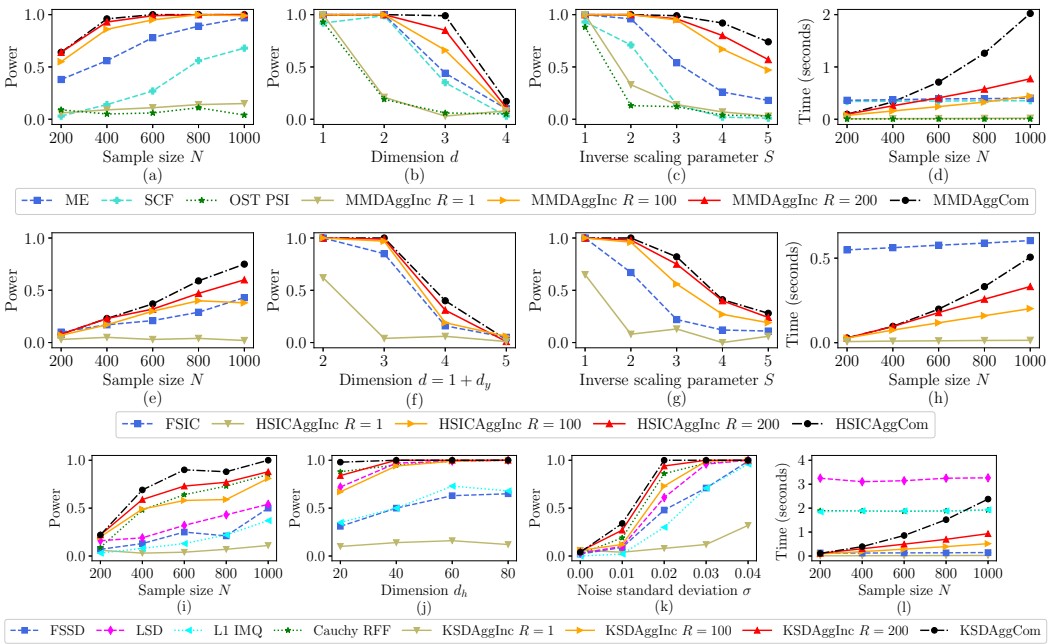

Figure 1: Two-sample *(a–d)* and independence *(e–h)* experiments using perturbed uniform densities. Goodness-of-fit *(i–l)* experiment using a Gaussian-Bernoulli Restricted Boltzmann Machine. The power results are averaged over 100 repetitions and the runtimes over 20 repetitions.

of the tests of Jitkrittum et al. (2016, 2017a,b) increase slower with the sample size than those of our aggregated tests with $R = 100, 200$, but a fixed computational cost is incurred for their optimization step, even for small sample sizes. In the goodness-of-fit framework, L1 IMQ performs similarly to FSSD which is in line with the results presented by Huggins and Mackey (2018, Figure 4d) who consider the same experiment. All other goodness-of-fit tests (except KSDAggInc $R = 1$) achieve much higher test power. Cauchy RFF and KSDAggInc $R = 200$ obtain similar power in almost all the experiments. While KSDAggInc $R = 200$ runs much faster in the experiments presented[6], it seems that the KSDAggInc runtimes increase more steeply with the sample size than the Cauchy RFF and L1 IMQ runtimes (see Appendix D.5 for details). LSD matches the power of KSDAggInc $R = 100$ when varying the noise level in Figure 1 (k) (KSDAggInc $R = 200$ has higher power), and when varying the hidden dimension in Figure 1 (j) where $d_x = 100$. When varying the sample size in Figure 1 (i), both KSDAggInc tests with $R = 100, 200$ achieve much higher power than LSD. Unsurprisingly, AggInc $R = 1$, which runs much faster than all the aforementioned tests, has low power in every experiment. For the two-sample problem, it obtains slightly higher power than OST PSI which runs even faster. We include more experiments in Appendix D: we present experiments on the MNIST dataset (same trends are observed), we use different collection of bandwidths, we verify that all tests have well-calibrated levels, and illustrate the benefits of the aggregation procedure.

## 9 Acknowledgements

Antonin Schrab acknowledges support from the U.K. Research and Innovation (EP/S021566/1). Ilmun Kim acknowledges support from the Yonsei University Research Fund of 2021-22-0332, and from the Basic Science Research Program through the National Research Foundation of Korea funded by the Ministry of Education (2022R1A4A1033384). Benjamin Guedj acknowledges partial support by the U.S. Army Research Laboratory and the U.S. Army Research Office, and by the U.K. Ministry of Defence and the U.K. Engineering and Physical Sciences Research Council (EP/R013616/1), and by the French National Agency for Research (ANR-18-CE40-0016-01 & ANR-18-CE23-0015-02). Arthur Gretton acknowledges support from the Gatsby Charitable Foundation.

---

[6]The runtimes in Figure 1 (d, h, l) can also vary due to the different implementations of the respective authors.

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
