# Supplementary material for 'Efficient Aggregated Kernel Tests using Incomplete $U$-statistics'

## A    Assumptions

### A.1    Assumptions for Theorem 1 and Theorem 3(i)

- $\max\left(\|p\|_\infty, \|q\|_\infty\right) \leq M$ for some $M > 0$
- $\alpha \in (0, e^{-1})$
- $\beta \in (0, 1)$
- $B_1 \geq 3\left(\log\left(8/\beta\right) + \alpha(1 - \alpha)\right)/\alpha^2$
- $\sigma_{2,\lambda}^2 \gtrsim 1$, where $\sigma_{2,\lambda}^2 := \mathbb{E}\left[h_\lambda(Z, Z')^2\right]$

### A.2    Assumptions for Theorem 2 and Theorem 3(ii)

- $\max\left(\|p\|_\infty, \|q\|_\infty\right) \leq M$ for some $M > 0$
- $\alpha \in (0, e^{-1})$
- $\beta \in (0, 1)$
- $B_1 \geq 12\left(\max_{\lambda \in \Lambda} w_\lambda^{-2}\right)\left(\log\left(8/\beta\right) + \alpha(1 - \alpha)\right)/\alpha^2$
- $B_2 \geq 8\log\left(2/\beta\right)/\alpha^2$
- $B_3 \geq \log_2\left(4\min_{\lambda \in \Lambda} w_\lambda^{-1}/\alpha\right)$
- $\sigma_{2,\lambda}^2 \gtrsim 1$ for $\lambda \in \Lambda$, where $\sigma_{2,\lambda}^2 := \mathbb{E}\left[h_\lambda(Z, Z')^2\right]$

## B    Background details on MMD, HSIC, KSD and quantile estimation

In this section, we present more background details than those presented in Section 2 on the Maximum Mean Discrepancy, on the Hilbert Schmidt Independence Criterion, and on the Kernel Stein Discrepancy.

**Maximum Mean Discrepancy.** Gretton et al. (2012a) introduce the *Maximum Mean Discrepancy* (MMD) which is a measure between probability densities $p$ and $q$ on $\mathbb{R}^d$. It is defined as the integral probability metric (IPM; Müller, 1997) over a reproducing kernel Hilbert space $\mathcal{H}_k$ (RKHS; Aronszajn, 1950) with associated kernel $k$. Gretton et al. (2012a, Lemma 4) show that the MMD is equal to the $\mathcal{H}_k$-norm of the difference between the mean embeddings $\mu_p(u) := \mathbb{E}_{X \sim p}\left[k(X, u)\right]$ and $\mu_q(u) := \mathbb{E}_{Y \sim q}\left[k(Y, u)\right]$ for $u \in \mathbb{R}^d$. The square of the MMD is equal to

$$\mathrm{MMD}_k^2(p, q) := \left(\sup_{f \in \mathcal{H}_k\,:\,\|f\|_{\mathcal{H}_k} \leq 1} \left|\mathbb{E}_p[f(X)] - \mathbb{E}_q[f(Y)]\right|\right)^2$$

$$= \|\mu_p - \mu_q\|_{\mathcal{H}_k}^2$$

$$= \mathbb{E}_{p,p}\left[k\left(X, X'\right)\right] - 2\,\mathbb{E}_{p,q}\left[k\left(X, Y\right)\right] + \mathbb{E}_{q,q}\left[k\left(Y, Y'\right)\right]$$

where $X$ and $X'$ (respectively $Y$ and $Y'$) are independent. Using a characteristic kernel (Fukumizu et al., 2008; Sriperumbudur et al., 2011) guarantees that $\mathrm{MMD}_k^2(p, q) = 0$ if and only if $p = q$, a crucial property for using the MMD to construct a two-sample test. With i.i.d. samples $\mathbb{X}_m := (X_i)_{1 \leq i \leq m}$ from $p$ and i.i.d. samples $\mathbb{Y}_n = (Y_j)_{1 \leq j \leq n}$ from $q$, Gretton et al. (2012a, Lemma 6) propose to use the unbiased quadratic-time MMD estimator $\widehat{\mathrm{MMD}}_k^2(\mathbb{X}_m, \mathbb{Y}_n)$ defined as

$$\frac{1}{m(m-1)} \sum_{(i,i') \in \mathbf{i}_2^m} k(X_i, X_{i'}) - \frac{2}{mn} \sum_{i=1}^m \sum_{j=1}^n k(X_i, Y_j) + \frac{1}{n(n-1)} \sum_{(j,j') \in \mathbf{i}_2^n} k(Y_j, Y_{j'})$$

$$= \frac{\mathbf{1}^\top \tilde{\mathbf{K}}_{\mathrm{XX}} \mathbf{1}}{m(m-1)} - 2\frac{\mathbf{1}^\top \mathbf{K}_{\mathrm{XY}} \mathbf{1}}{mn} + \frac{\mathbf{1}^\top \tilde{\mathbf{K}}_{\mathrm{YY}} \mathbf{1}}{n(n-1)}$$

where $\tilde{\mathbf{K}}_{\mathrm{XX}}$ and $\tilde{\mathbf{K}}_{\mathrm{YY}}$ are the kernel matrices $\mathbf{K}_{\mathrm{XX}} := \big(k(X_i, X_j)\big)_{1 \le i,j \le m}$ and $\mathbf{K}_{\mathrm{YY}} := \big(k(Y_i, Y_j)\big)_{1 \le i,j \le n}$ with diagonal entries set to 0, where $\mathbf{K}_{\mathrm{XY}} := \big(k(X_i, Y_j)\big)_{1 \le i \le m, 1 \le j \le n}$, and where $\mathbf{1}$ is a one-dimensional vector with all entries equal to 1 of variable length determined by the context[7]. As noted by Kim et al. (2022), this MMD estimator can be rewritten as a two-sample $U$-statistic (both of second order) (Hoeffding, 1992)

$$\widehat{\mathrm{MMD}}_k^2(\mathbb{X}_m, \mathbb{Y}_n) = \frac{1}{|\mathbf{i}_2^m||\mathbf{i}_2^n|} \sum_{(i,i') \in \mathbf{i}_2^m} \sum_{(j,j') \in \mathbf{i}_2^n} h_k^{\mathrm{MMD}}(X_i, X_{i'}; Y_j, Y_{j'})$$

where $\mathbf{i}_a^b$ denotes the set of all $a$-tuples drawn without replacement from $\{1, \ldots, b\}$ so that $|\mathbf{i}_a^b| = b \cdots (b - a + 1)$, for example $|\mathbf{i}_2^m| = m(m-1)$, and where, for $x_1, x_2, y_1, y_2 \in \mathbb{R}^d$, we let

$$h_k^{\mathrm{MMD}}(x_1, x_2; y_1, y_2) := k(x_1, x_2) - k(x_1, y_2) - k(x_2, y_1) + k(y_1, y_2).$$

This kernel can easily be symmetrized (Kim et al., 2022) using a symmetrization trick (Duembgen, 1998), this corresponds to working with

$$\bar{h}_k^{\mathrm{MMD}}(x_1, x_2; y_1, y_2) := \frac{1}{2!2!} \sum_{(i_1,i_2) \in \mathbf{i}_2^2} \sum_{(j_1,j_2) \in \mathbf{i}_2^2} h_k^{\mathrm{MMD}}(x_1, x_2; y_1, y_2)$$

and the MMD expression as a $U$-statistic still holds when replacing $h_k^{\mathrm{MMD}}$ with its symmetrized variant $\bar{h}_k^{\mathrm{MMD}}$.

**Hilbert Schmidt Independence Criterion.** For a joint probability density $p_{xy}$ on $\mathbb{R}^{d_x} \times \mathbb{R}^{d_y}$ with marginals $p_x$ on $\mathbb{R}^{d_x}$ and $p_y$ on $\mathbb{R}^{d_y}$, Gretton et al. (2005) introduce the *Hilbert Schmidt Independence Criterion* (HSIC) which is defined as

$$\mathrm{HSIC}_{k,\ell}(p_{xy}) := \mathrm{MMD}_\kappa^2(p_{xy}, p_x p_y)$$
$$= \mathbb{E}_{p_{xy}, p_{xy}}\Big[k(X, X')\ell(Y, Y')\Big] - 2\,\mathbb{E}_{p_{xy}}\Big[\mathbb{E}_{p_x}[k(X, X')]\mathbb{E}_{p_y}[\ell(Y, Y')]\Big]$$
$$+ \mathbb{E}_{p_x, p_x}\Big[k(X, X')\Big]\mathbb{E}_{p_y, p_y}\Big[k(Y, Y')\Big].$$

with kernels $k$ on $\mathbb{R}^{d_x}$ and $\ell$ on $\mathbb{R}^{d_y}$ giving the product kernel $\kappa\big((x, y), (x', y')\big) := k(x, x')\ell(y, y')$ on $\mathbb{R}^{d_x} \times \mathbb{R}^{d_y}$. With i.i.d. pairs of samples $\mathbb{Z}_N := (Z_i)_{1 \le i \le N} = \big((X_i, Y_i)\big)_{1 \le i \le N}$ drawn from $p_{xy}$, a natural unbiased HSIC estimator (Gretton et al., 2008; Song et al., 2012) is then

$$\widehat{\mathrm{HSIC}}_{k,\ell}(\mathbb{Z}_N) := \frac{1}{|\mathbf{i}_2^N|} \sum_{(i,j) \in \mathbf{i}_2^N} k(X_i, X_j)\ell(Y_i, Y_j) - \frac{2}{|\mathbf{i}_3^N|} \sum_{(i,j,r) \in \mathbf{i}_3^N} k(X_i, X_j)\ell(Y_i, Y_r)$$
$$+ \frac{1}{|\mathbf{i}_4^N|} \sum_{(i,j,r,s) \in \mathbf{i}_4^N} k(X_i, X_j)\ell(Y_r, Y_s)$$
$$= \frac{1}{|\mathbf{i}_4^N|} \sum_{(i,j,r,s) \in \mathbf{i}_4^N} h_{k,\ell}^{\mathrm{HSIC}}(Z_i, Z_j, Z_r, Z_s)$$

which is a fourth-order one-sample $U$-statistic. For $z_a = (x_a, y_a) \in \mathbb{R}^{d_x} \times \mathbb{R}^{d_y}$, $a = 1, \ldots, 4$, we let

$$h_{k,\ell}^{\mathrm{HSIC}}(z_1, z_2, z_3, z_4) := \frac{1}{4} h_k^{\mathrm{MMD}}(x_1, x_2; x_3, x_4) h_\ell^{\mathrm{MMD}}(y_1, y_2; y_3, y_4).$$

We stress the fact that this HSIC estimator can actually be computed in quadratic time as shown by Song et al. (2012, Equation 5) who provide the following closed-form expression

$$\widehat{\mathrm{HSIC}}_{k,\ell}(\mathbb{Z}_N) = \frac{1}{N(N-3)} \left( \mathrm{tr}(\tilde{\mathbf{K}}\tilde{\mathbf{L}}) + \frac{\mathbf{1}^\top \tilde{\mathbf{K}} \mathbf{1}\mathbf{1}^\top \tilde{\mathbf{L}} \mathbf{1}}{(N-1)(N-2)} - \frac{2}{N-2} \mathbf{1}^\top \tilde{\mathbf{K}}\tilde{\mathbf{L}} \mathbf{1} \right)$$

where $\tilde{\mathbf{K}}$ and $\tilde{\mathbf{L}}$ are the kernel matrices $\mathbf{K} := \big(k(X_i, X_j)\big)_{1 \le i,j \le N}$ and $\mathbf{L} := \big(\ell(Y_i, Y_j)\big)_{1 \le i,j \le N}$ with diagonal entries set to 0. Again, this kernel can be symmetrized (Song et al., 2012; Kim et al., 2022)

---

[7]We use this convention for the notation $\mathbf{1}$ in this whole section.

using a symmetrization trick (Duembgen, 1998), and the HSIC expression as a $U$-statistic still holds when replacing $h_k^{\mathrm{HSIC}}$ with its symmetrized variant

$$\bar{h}_k^{\mathrm{HSIC}}(z_1, z_2, z_3, z_4) \coloneqq \frac{1}{4!} \sum_{(i_1, i_2, i_3, i_4) \in \mathbf{i}_4^4} h_k^{\mathrm{HSIC}}(z_{i_1}, z_{i_2}, z_{i_3}, z_{i_4}).$$

**Kernel Stein Discrepancy.** For probability densities $p$ and $q$ on $\mathbb{R}^d$, Chwialkowski et al. (2016) and Liu et al. (2016) introduce the *Kernel Stein Discrepancy* (KSD) defined as

$$\begin{aligned}
\mathrm{KSD}_{p,k}^2(q) &\coloneqq \mathrm{MMD}_{h_{k,p}^{\mathrm{KSD}}}^2(q, p) \\
&= \mathbb{E}_{q,q}\big[h_{k,p}^{\mathrm{KSD}}(Z, Z')\big] - 2\,\mathbb{E}_{q,p}\big[h_{k,p}^{\mathrm{KSD}}(Z, X)\big] + \mathbb{E}_{p,p}\big[h_{k,p}^{\mathrm{KSD}}(X, X')\big] \\
&= \mathbb{E}_{q,q}\big[h_{k,p}^{\mathrm{KSD}}(Z, Z')\big]
\end{aligned}$$

where the *Stein kernel* $h_{p,k} \colon \mathbb{R}^d \times \mathbb{R}^d \to \mathbb{R}$ is defined as

$$\begin{aligned}
h_{k,p}^{\mathrm{KSD}}(x, y) &\coloneqq \big(\nabla \log p(x)^\top \nabla \log p(y)\big)\, k(x, y) + \nabla \log p(y)^\top \nabla_x k(x, y) \\
&\quad + \nabla \log p(x)^\top \nabla_y k(x, y) + \sum_{i=1}^d \frac{\partial}{\partial x_i \partial y_i}\, k(x, y).
\end{aligned}$$

The Stein kernel satisfies the Stein identity $\mathbb{E}_p[h_{k,p}^{\mathrm{KSD}}(X, \cdot)] = 0$. The KSD is particularly useful for the goodness-of-fit setting with a model density $p$ and i.i.d. samples $\mathbb{Z}_N \coloneqq (Z_i)_{1 \le i \le N}$ drawn from a density $q$ because it admits an estimator which does not require samples from the model $p$. The quadratic-time KSD estimator can be computed as the second-order one-sample $U$-statistic

$$\widehat{\mathrm{KSD}}_{p,k}^2(\mathbb{Z}_N) \coloneqq \frac{1}{|\mathbf{i}_2^N|} \sum_{(i,j) \in \mathbf{i}_2^N} h_{k,p}^{\mathrm{KSD}}(Z_i, Z_j) = \frac{\mathbf{1}^\top \tilde{\mathbf{H}} \mathbf{1}}{N(N-1)}$$

where $\tilde{\mathbf{H}}$ is the kernel matrix $\mathbf{H} \coloneqq \big(h_{k,p}^{\mathrm{KSD}}(Z_i, Z_j)\big)_{1 \le i,j \le N}$ with diagonal entries set to 0. The Stein kernel $h_k^{\mathrm{KSD}}$ is already symmetric, we can write $\bar{h}_k^{\mathrm{KSD}}(x, y) \coloneqq h_k^{\mathrm{KSD}}(x, y)$ for all $x, y \in \mathbb{R}^d$ for consistency of notation. As presented in Section 2, Chwialkowski et al. (2016, Theorem 2.2) show the consistency of the KSD goodness-of-fit provided that the kernel $k$ is $C_0$-universal (Carmeli et al., 2010, Definition 4.1) and that

$$\mathbb{E}_q\Big[h_{k,p}^{\mathrm{KSD}}(z, z)\Big] < \infty \qquad \text{and} \qquad \mathbb{E}_q\left[\left\|\nabla \log\left(\frac{p(z)}{q(z)}\right)\right\|_2^2\right] < \infty.$$

as introduced in Equation (7).

**Quantile estimation.** There exist many approaches to estimate the quantiles of the test statistics under the null hypothesis in the three frameworks: using the quantile of a known distribution-free asymptotic null distribution (Gretton et al., 2008, 2012a), sampling from an asymptotic null distribution with eigenspectrum approximation (Gretton et al., 2009), using permutations (Gretton et al., 2008; Albert et al., 2022; Kim et al., 2022; Schrab et al., 2021), using a wild bootstrap (Fromont et al., 2012; Chwialkowski et al., 2014, 2016; Schrab et al., 2021, 2022), using a parametric bootstrap (Key et al., 2021; Schrab et al., 2022), using other bootstrap methods (Liu et al., 2016), to name but a few. Permutation-based tests have been shown to correctly control the non-asymptotic level for the two-sample (Schrab et al., 2021; Kim et al., 2022) and independence (Albert et al., 2022; Kim et al., 2022) problems. For the two-sample test, using a wild bootstrap also guarantees well-calibrated non-asymptotic level (Fromont et al., 2012; Schrab et al., 2021). For the goodness-of-fit setting, while a wild bootstrap guarantees only control of the asymptotic level (Chwialkowski et al., 2016), using a parametric bootstrap results in a well-calibrated non-asymptotic level (Schrab et al., 2022). In this work, we focus on the wild bootstrap approach, though we point out that our results also hold using a parametric bootstrap for the goodness-of-fit setting as done by Schrab et al. (2022).

## C Detailed experimental protocol

In this section, we present details on our experiments and on the tests considered.

**Implementation and computational resources.** All experiments have been run on an AMD Ryzen Threadripper 3960X 24 Cores 128Gb RAM CPU at 3.8GHz, except the LSD test (Grathwohl et al., 2020) for which a neural network has been trained using an NVIDIA RTX A5000 24Gb Graphics Card. The overall runtime of all the experiments is of the order of a couple of hours (significant speedup can be obtained by using parallel computing). We use the implementations of the respective authors (all under the MIT license) for the ME, SCF, FSIC and FSSD tests of Jitkrittum et al. (2016, 2017a,b), for the LSD test of Grathwohl et al. (2020), for the L1 IMQ and Cauchy RFF tests of Huggins and Mackey (2018), and for the OST PSI test of Kübler et al. (2020). The implementation of our computationally efficient aggregated tests, as well as the code for reproducibility of the experiments, are available here under the MIT license.

**Kernels.** For the two-sample and independence experiments, we use the Gaussian kernel[8] with equal bandwidths $\lambda_1 = \cdots = \lambda_d = \tilde{\lambda}$, which is defined as

$$k_\lambda(x, y) := \exp\left(-\sum_{i=1}^{d} \frac{(x_i - y_i)^2}{\lambda_i^2}\right) = \exp\left(-\frac{\|x - y\|_2^2}{\tilde{\lambda}^2}\right),$$

and similarly for the kernel $\ell_\mu$. As shown by Gorham and Mackey (2017), a more appropriate kernel for goodness-of-fit testing is the IMQ (inverse multiquadric) kernel

$$k_\lambda(x, y) := \left(1 + \sum_{i=1}^{d} \frac{(x_i - y_i)^2}{\lambda_i^2}\right)^{-\beta_k} = \left(1 + \frac{\|x - y\|_2^2}{\tilde{\lambda}^2}\right)^{-\beta_k} \propto \left(\tilde{\lambda}^2 + \|x - y\|_2^2\right)^{-\beta_k} \quad (18)$$

for some $\beta_k \in (0, 1)$. In our goodness-of-fit experiments, we use the IMQ kernel with fixed parameter $\beta_k = 0.5$.

**Two-sample and independence experiments.** In our experiments, we consider perturbed uniform densities, those can be shown to lie in Sobolev balls and are used to derive the minimax rates over Sobolev balls for the two-sample and independence problems (Li and Yuan, 2019; Albert et al., 2022). For the two-sample problem, we consider testing samples drawn from a uniform density against samples drawn from a perturbed uniform density, as considered by Schrab et al. (2021, see Equation 17 for formal definition and Figure 2 for illustrations). We scale the perturbations so that the perturbed density takes value in the whole interval $[0, 2]$, we then consider some inverse scaling parameter $S \geq 1$ such that it takes value in the interval $[1 - 1/S, 1 + 1/S]$. Intuitively, as $S$ increases, the perturbation is shrunk. In Figure 1 (a, d), we consider 2 perturbations with inverse scaling parameter $S = 2$ in dimension $d = 1$ and vary the sample size $N \in \{200, 400, 600, 800, 1000\}$. In Figure 1 (b), we vary the dimension $d \in \{1, 2, 3, 4\}$ for 1 perturbation with $S = 1$ and $N = 1000$. In Figure 1 (c), we use 1 perturbation with $d = 1$ and $N = 1000$, we vary the inverse scaling parameter $S \in \{1, 2, 3, 4, 5\}$. For the independence problem, we draw samples from the joint perturbed uniform density in dimension $d_x + d_y$, the marginals are simply uniform densities in dimensions $d_x$ and $d_y$, respectively. We fix $d_x = 1$ and vary $d_y$ exactly as in the two-sample setting. The parameters for the independence experiments in Figure 1 (e–h) are the same as those of the two-sample experiments in Figure 1 (a–d) detailed above (with the only difference that for Figure 1 (f) we consider $d_y \in \{1, 2, 3\}$).

**Goodness-of-fit experiments.** In Figure 1 (i–l), we use a Gaussian-Bernoulli Restricted Boltzmann Machine (GBRBM) with the same setting considered by Liu et al. (2016), Grathwohl et al. (2020) and Schrab et al. (2022). This is a hidden variable model with a continuous observable variable in $\mathbb{R}^{d_x}$ and a hidden binary variable in $\{-1, 1\}^{d_h}$, the joint density is intractable but the score function admits a closed form. The GBRBM has parameters $b \in \mathbb{R}^{d_x}$ and $c \in \mathbb{R}^{d_h}$, which are drawn from Gaussian standard distributions, and a matrix parameter $B \in \mathbb{R}^{d_x \times d_h}$. For the model $p$, the elements of $B$ are sampled uniformly from $\{-1, 1\}$ (i.i.d. Rademacher variables). The samples come from a GBRBM $q$ with the same parameters as the model $p$ but where some Gaussian noise $\mathcal{N}(0, \sigma)$ is injected into the elements of $B$. In Figure 1 (i, l), we consider dimensions $d_x = 50$ and $d_h = 40$ with noise standard deviation $\sigma = 0.02$ and we vary the sample size $N \in \{200, 400, 600, 800, 1000\}$. In Figure 1 (j), we fix $d_x = 100$, $N = 1000$, $\sigma = 0.03$ and we vary the hidden dimension $d_h \in \{20, 40, 60, 80\}$. For fixed observed dimension $d_x$, as the hidden dimension $d_h$ increases the size of $B \in \mathbb{R}^{d_x \times d_h}$ becomes larger, so there is more evidence of the noise being injected, which makes the problem easier. Hence, the test power increases as $d_h$ increases for fixed $d_x$. In Figure 1 (k), we consider

---

[8]In practice, we do not need to normalize the kernels to integrate to 1 since our tests are invariant to multiplying the kernel by a scalar.

dimensions $d_x = 50$ and $d_h = 40$ with sample size $N = 1000$, we vary the noise standard deviations $\sigma \in \{0, 0.01, 0.02, 0.03, 0.04\}$.

**AggInc tests.** As in Schrab et al. (2021), for MMDAggInc, we use a collection of $B = 10$ bandwidths defined as

$$\left\{ (4\lambda_{\max}/\lambda_{\min})^{(i-1)/(B-1)} : i = 1, \ldots, B \right\}$$

which is a discretisation of the interval $[\lambda_{\min}/2, 2\lambda_{\max}]$ where $\lambda_{\min}$ and $\lambda_{\max}$ are the minimal and maximal inter-sample distances, respectively. If the minimal distance is smaller than $10^{-1}$, we consider the 5% smallest inter-sample positive distance instead, if it is still smaller than $10^{-1}$ we set $\lambda_{\min} = 10^{-1}$.

Similarly to Schrab et al. (2022), for KSDAggInc, we use a collection of $B = 10$ bandwidths defined as

$$\left\{ d^{-1} \lambda_{\max}^{(i-1)/(B-1)} : i = 1, \ldots, B \right\}$$

where $d$ is the dimension of the samples, and where $\lambda_{\max} = \max\left\{ \|z_i - z_j\|_2 : (i,j) \in \mathbf{i}_2^N \right\}$ is the maximal inter-sample distance (which is thresholded at 2 if it is smaller than this value). This collection discretises the interval $\left[ d^{-1}, d^{-1}\lambda_{\max} \right]$.

For HSICAggInc, we work with the collection of 25 pairs of bandwidths

$$\Lambda := \left\{ \left( 2^i \lambda_{\text{med}} \mathbf{1}_{d_x}, 2^j \mu_{\text{med}} \mathbf{1}_{d_y} \right) : i, j \in \{-2, -1, 0, 1, 2\} \right\}$$

for the kernels $k_\lambda$ and $\ell_\mu$ defined in Equation (13), where

$$\lambda_{\text{med}} := \text{median}\left\{ \|x_i - x_j\|_2 : (i,j) \in \mathbf{i}_2^N \right\} \quad \text{and} \quad \mu_{\text{med}} := \text{median}\left\{ \|y_i - y_j\|_2 : (i,j) \in \mathbf{i}_2^N \right\}.$$

We also consider collections of this form for MMDAggInc and KSDAggInc in Appendix D.2 but this results in a small loss of power.

In practice, depending on our computational budget, we can also consider multiple kernels, each with various bandwidths, as considered in Schrab et al. (2021). All aggregated tests are run with uniform weights defined as $w_\lambda := 1/|\Lambda|$ for all $\lambda \in \Lambda$. The design choice consists of $R$ sub-diagonals of the kernel matrix for $R \in \{1, 100, 200\}$, it is formally defined in Section 8. We also consider the quadratic-time case where the full design is considered (*i.e.* case $R = N - 1$), we refer to these tests using complete $U$-statistics as AggCom for consistency. We note that MMDAggCom, HSICAggCom and KSDAggCom simply correspond to the quadratic-time MMDAgg, HSICAgg and KSDAgg tests proposed by Schrab et al. (2021), Albert et al. (2022) and Schrab et al. (2022), respectively, with the only difference being their implementation: Agg tests run slightly faster than AggCom tests since they can exploit the fact that the whole kernel matrix needs to be computed. We use $B_1 = 500$ and $B_2 = 500$ wild bootstrapped statistics to estimate the quantiles and the probability under the null for the correction in Equation (17), respectively. In practice, we recommend using either $B_1 = B_2 = 500$ for having fast tests, or $B_1 = B_2 = 2000$ for obtaining slightly higher power (with longer runtimes). For that correction term, we use $B_3 = 50$ steps of bisection method to approximate the supremum.

**ME, SCF, FSIC and FSSD tests.** Jitkrittum et al. (2016) use the two-sample tests ME and SCF proposed by Chwialkowski et al. (2015) with features which are chosen to maximise a lower bound on the test power. The ME test is based on analytic Mean Embeddings while the SCF test uses the difference in Smooth Characteristic Functions. For the independence problem, Jitkrittum et al. (2017a) construct a FSIC test which uses their proposed normalised Finite Set Independence Criterion. Jitkrittum et al. (2017b) propose a goodness-of-fit test based on the Finite Set Stein Discrepancy (FSSD). All those tests utilise test statistics which evaluate the witness function of either the MMD, HSIC, or KSD, at some test locations (*i.e.* features) chosen on held-out data to maximise test power. For the two-sample SCF test, the test locations are in the frequency domain rather than in the spatial domain. All tests are used with 10 test locations which are chosen on half of the data, as done in the experiments of Jitkrittum et al. (2016, 2017a,b). The ME and SCF tests use the quantiles of their known chi-square asymptotic null distributions. The FSIC test uses 500 permutations to simulate the null hypothesis and compute the test threshold to ensure a well-calibrated non-asymptotic level. The FSSD test simulates 2000 samples from the asymptotic null distribution (weighted sum of chi-squares) with the eigenvalues being computed from the covariance matrix with respect to the observed samples. For the two-sample and independence tests, the bandwidths of the Gaussian

kernels are selected during the optimization procedure. For the goodness-of-fit test, the bandwidth of the IMQ (inverse multiquadric) kernel is set to some fixed value as done by Jitkrittum et al. (2017b), following the recommendation of Gorham and Mackey (2017).

**LSD test.** The Kernelised Stein Discrepancy (KSD) is a Stein Discrepancy (Gorham and Mackey, 2017) where the class of functions is taken to be the unit ball of a reproducing kernel Hilbert space (RKHS). Grathwohl et al. (2020) propose to instead consider some more expressive class of functions consisting of neural networks, resulting in the Learned Stein Discrepancy (LSD). For goodness-of-fit testing, they propose to split the data into training (80%), validation (10%) and testing (10%) sets. They construct a test statistic which is asymptotically normal under both $\mathcal{H}_0$ and $\mathcal{H}_1$. Using the training set, they train the parametrised neural network to maximise the test power by optimizing a proxy for it which is derived following the reasonings of Gretton et al. (2012b), Sutherland et al. (2017) and Jitkrittum et al. (2017b). They perform model selection on the validation set. Finally, they run the test on the testing set using the quantile of the asymptotic normal distribution under the null. As in the experiments of Grathwohl et al. (2020), a 2-layer MLP with 300 units per layer and with Swish nonlinearity (Ramachandran et al., 2017) is used. Their model is trained using the Adam optimizer (Kingma and Ba, 2014) for 1000 iterations, with dropout (Srivastava et al., 2014), with weight decay of strength 0.0005, with learning rate $10^{-3}$, and with $L^2$ regularising strength 0.5.

**Cauchy RFF and L1 IMQ tests.** Huggins and Mackey (2018) introduce random feature Stein discrepancies (RΦSDs) which are computable in linear time. The FSSD of Jitkrittum et al. (2017b) corresponds to a specific RΦSD. Another special case of their general RΦSDs is the random Fourier feature (RFF; Rahimi and Recht, 2007) approximation of KSD. They consider in their experiments both Gaussian and Cauchy RFF tests, they observe that Cauchy RFF significantly outperforms its Gaussian counterpart (Huggins and Mackey, 2018, Figure 4). Using the inverse multiquadric kernel (IMQ; Equation (18)), for which Gorham and Mackey (2017) showed that KSD dominates weak convergence when $\beta_k \in (0, 1)$, Huggins and Mackey (2018, Example 3.4) derive a $L^r$ IMQ RΦSD, with some simple setting when $r = 1$. They show in their experiments that L1 IMQ has superior performance compared to all other tests considered for experiments comparing Gaussian and Laplace distributions, as well as Gaussian and multivariate $t$ distributions. We use the parameters recommended by the authors when running Cauchy RFF and L1 IMQ, except for the number of samples drawn from the unnormalized density to estimate the covariance matrix to simulate the null hypothesis. As explained in Appendix D.5, we tune that number in order for their tests to be more computationally efficient while retaining their high test power.

**OST PSI test.** Kübler et al. (2020) construct an MMD adaptive two-sample test which exploits the post-selection inference framework (PSI; Fithian et al., 2014; Lee et al., 2016) (with uncountable candidate sets) to use the same data to both perform kernel selection and run the test while still guaranteeing control of the probability of type I error. Their one-sided test (OST) runs in linear time and does not rely on data splitting. For kernel selection, they use a proxy for asymptotic power as a criterion. We use their implementation with the same collection of bandwidths as for our MMDAggInc test as specified above.

# D   Additional experiments

In this section, we present additional experiments. We consider more challenging experiments on the high-dimensional MNIST dataset. We report results using different collections of bandwidths. We empirically show that all the tests considered have well-calibrated levels. We present experiments highlighting the strengths of the aggregation procedure. Finally, we discuss the choice of parameters for the L1 IMQ and Cauchy RFF tests.

## D.1   MNIST Experiments

In Figure 2, we run experiments on the real-world MNIST dataset (LeCun et al., 2010) consisting of images of digits in dimension 784.

For the two-sample problem, the distribution $P$ consists of images of all digits and the other distribution is $Q_i$ where $Q_1$ consists of images of only the five odd digits, $Q_2$ is $Q_1$ with 0, $Q_3$ is $Q_2$ with 2, $Q_4$ is $Q_3$ with 4, $Q_5$ is $Q_4$ with 6 (*i.e.* $Q_5$ consists of images of all digits expect 8). This setting has previously been considered by Schrab et al. (2021).

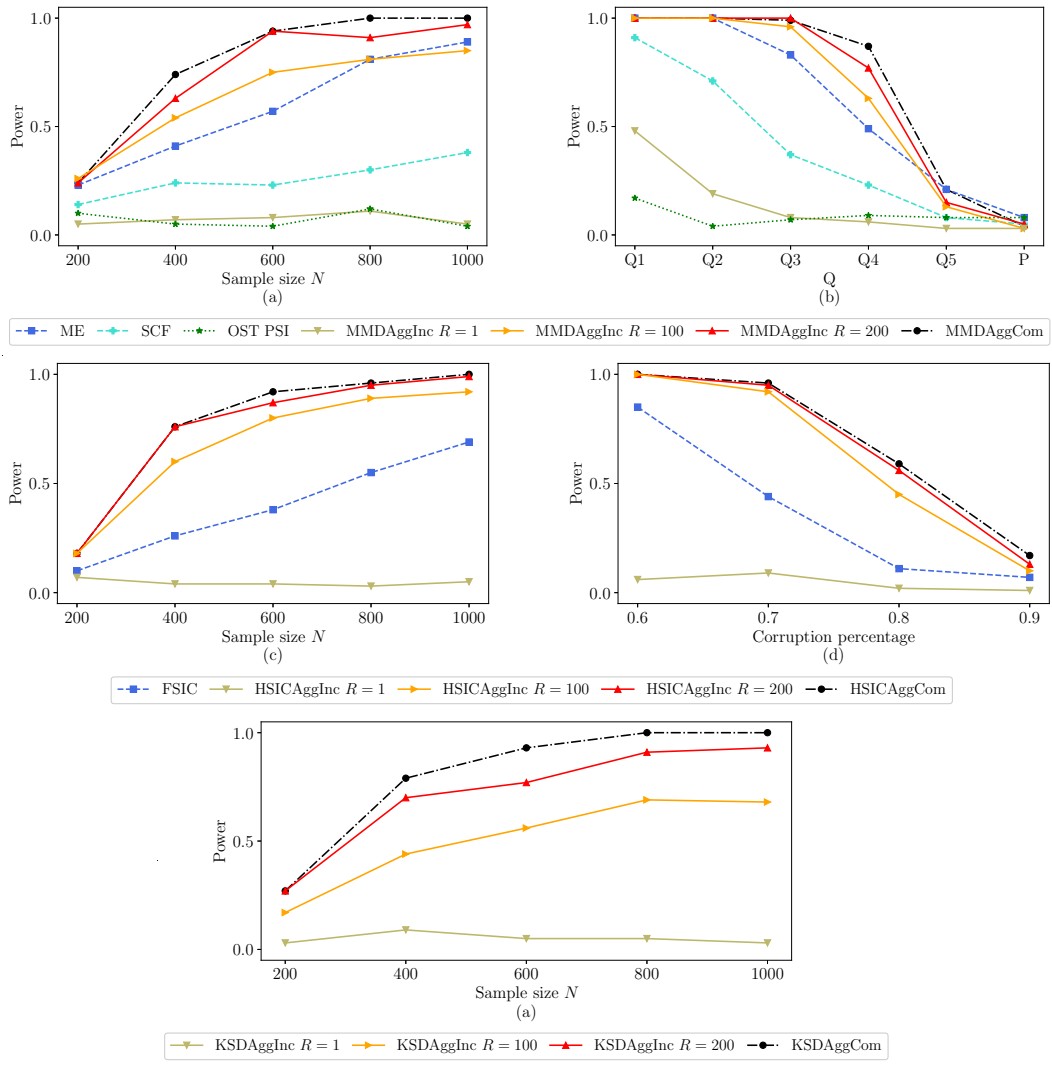

Figure 2: Experiments using the MNIST dataset for the *(a–b)* two-sample, *(c–d)* independence, and *(e)* goodness-of-fit problems. The power results are averaged over 100 repetitions.

For the independence problem, we pair each image of a digit with the value of the digit. To make the problem more challenging, we corrupt some percentage of the data by pairing images with values of random digits.

For the goodness-of-fit problem, the samples are drawn from the true MNIST dataset and the model is a Normalizing Flow (generative model which admits a density; Dinh et al., 2017; Kingma and Dhariwal, 2018) trained on the MNIST dataset. Since we have access only to pre-computed values of the score function evaluated at some MNIST samples but do not have access to the score function itself, we found that computing FSSD, L1 IMQ or Cauchy RFF to be very challenging; for this reason the results for those tests are not reported.

Overall, we observe the same trends in Figure 2 for this high-dimensional real-world setting as we did in Figure 1 in the lower-dimensional setting in which the Sobolev smoothness assumption is satisfied for MMDAggInc and HSICAggInc. Indeed, the AggInc $R = 200$ tests clearly outperform the tests we compare against and even match the power of AggCom in several experiments. ME and SCF obtain significantly lower power than MMDAggInc $R = 100$ in various settings in Figure 2 (a, b). We observe that HSICAggInc $R = 100$ significantly outperforms FSIC in both independence experiments in Figure 2 (c, d). For the goodness-of-fit setting, the tests manage to detect that the true MNIST samples are not drawn from the density of the trained Normalizing Flow. There is

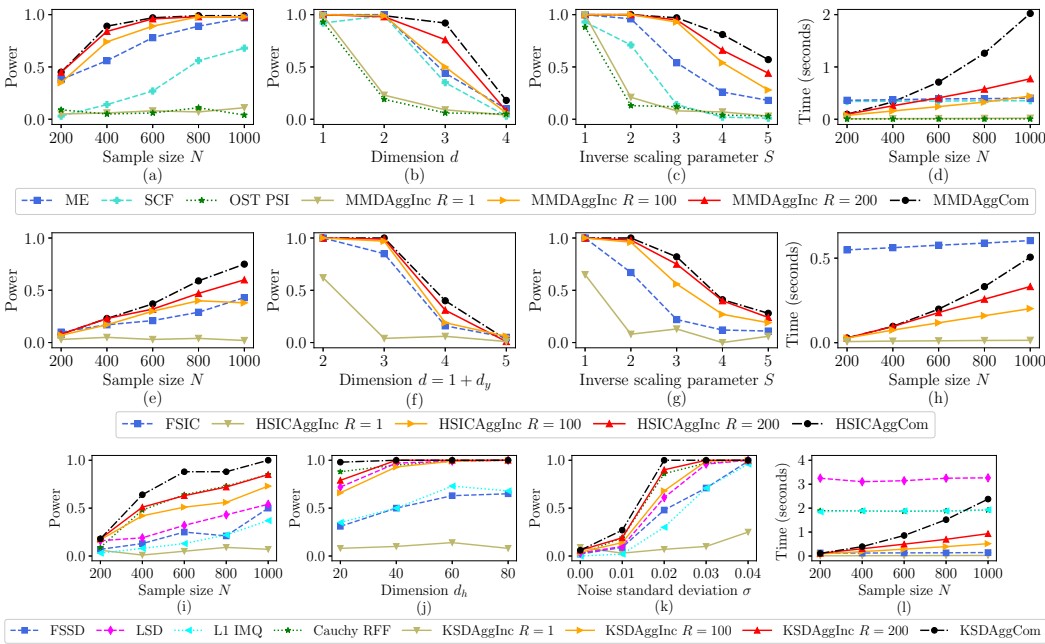

Figure 3: Two-sample *(a–d)* and independence *(e–h)* experiments using perturbed uniform densities. Goodness-of-fit *(i–l)* experiment using a Gaussian-Bernoulli Restricted Boltzmann Machine. The power results are averaged over 100 repetitions and the runtimes over 20 repetitions.

a significant power difference between each of the four tests: KSDAggInc $R = 1, 100, 200$ and KSDAggCom.

## D.2 Different collections for MMDAggInc and KSDAggInc

In Figure 3, we reproduce the experiments presented in Figure 1 using, for MMDAggInc and KSDAggInc, the collection of 21 bandwidths

$$\Lambda := \left\{ 2^i \lambda_{\mathrm{med}} \mathbf{1}_d : i \in \{-10, \dots, 10\} \right\} \quad \text{where} \quad \lambda_{\mathrm{med}} := \mathrm{median} \left\{ \|z_i - z_j\|_2 : (i, j) \in \mathbf{i}_2^N \right\},$$

where $\mathbf{1}_d$ is a $d$-dimensional vector with all entries equal to 1. We observe that using this collection leads to slightly lower power for MMDAggInc and KSDAggInc than in Figure 1 with different collections. In Figure 3, KSDAggInc $R = 200$ obtains exactly the same power as Cauchy RFF. The results for HSICAggInc in Figure 3 are the same as those of Figure 1, we simply report them for consistency.

## D.3 Well-calibrated levels

All tests are run with level $\alpha = 0.05$, it is verified in Tables 1 to 6 that all tests have well-calibrated levels for the three testing frameworks, when varying either the sample size or the dimension. The levels plotted are averages obtained across 200 repetitions, this explains the small fluctuations observed from the desired test level $\alpha = 0.05$. The settings of those six experiments correspond to the settings of the experiments presented in Figure 1 (a, b, e, f, i, j) detailed above, with the difference that we are working under the null hypothesis (*i.e.* perturbed uniform densities are replaced with uniform densities, and the noise standard deviation for the Gaussian-Bernoulli Restricted Boltzmann Machine is set to $\sigma = 0$).

Table 1: Two-sample level experiment using uniform densities varying the sample size.

| Sample size | ME | SCF | OST PSI | MMDAggInc $R = 1$ | MMDAggInc $R = 100$ | MMDAggInc $R = 200$ | MMDAggCom |
|---|---|---|---|---|---|---|---|
| 200 | 0.055 | 0.005 | 0.045 | 0.04 | 0.05 | 0.055 | 0.055 |
| 400 | 0.08 | 0.01 | 0.04 | 0.035 | 0.06 | 0.03 | 0.03 |
| 600 | 0.08 | 0.005 | 0.105 | 0.085 | 0.04 | 0.04 | 0.07 |
| 800 | 0.05 | 0.005 | 0.055 | 0.075 | 0.03 | 0.035 | 0.055 |
| 1000 | 0.075 | 0.005 | 0.045 | 0.045 | 0.015 | 0.02 | 0.05 |

Table 2: Two-sample level experiment using uniform densities varying the dimension.

| Dimension | ME | SCF | OST PSI | MMDAggInc $R = 1$ | MMDAggInc $R = 100$ | MMDAggInc $R = 200$ | MMDAggCom |
|---|---|---|---|---|---|---|---|
| 1 | 0.045 | 0 | 0.035 | 0.02 | 0.045 | 0.04 | 0.045 |
| 2 | 0.045 | 0.035 | 0.085 | 0.1 | 0.05 | 0.04 | 0.035 |
| 3 | 0.04 | 0.05 | 0.04 | 0.04 | 0.05 | 0.06 | 0.025 |
| 4 | 0.045 | 0.05 | 0.03 | 0.055 | 0.045 | 0.045 | 0.03 |

Table 3: Independence level experiment using uniform densities varying the sample size.

| Sample size | FSIC | HSICAggInc $R = 1$ | HSICAggInc $R = 100$ | HSICAggInc $R = 200$ | HSICAggCom |
|---|---|---|---|---|---|
| 200 | 0.04 | 0.055 | 0.035 | 0.035 | 0.035 |
| 400 | 0.045 | 0.05 | 0.04 | 0.05 | 0.05 |
| 600 | 0.05 | 0.035 | 0.05 | 0.06 | 0.05 |
| 800 | 0.03 | 0.07 | 0.02 | 0.035 | 0.04 |
| 1000 | 0.07 | 0.02 | 0.085 | 0.035 | 0.04 |

Table 4: Independence level experiment using uniform densities varying the dimension.

| Dimension | FSIC | HSICAggInc $R = 1$ | HSICAggInc $R = 100$ | HSICAggInc $R = 200$ | HSICAggCom |
|---|---|---|---|---|---|
| 2 | 0.035 | 0.065 | 0.08 | 0.055 | 0.07 |
| 3 | 0.065 | 0.055 | 0.035 | 0.02 | 0.025 |
| 4 | 0.04 | 0.035 | 0.045 | 0.055 | 0.055 |

Table 5: Goodness-of-fit level experiment using a Gaussian-Bernoulli Restricted Boltzmann Machine varying the sample size.

| Sample size | FSSD | LSD | KSDAggInc $R = 1$ | KSDAggInc $R = 100$ | KSDAggInc $R = 200$ | KSDAggCom |
|---|---|---|---|---|---|---|
| 200 | 0.02 | 0.07 | 0.05 | 0.045 | 0.06 | 0.06 |
| 400 | 0.03 | 0.04 | 0.06 | 0.04 | 0.065 | 0.055 |
| 600 | 0.04 | 0.075 | 0.03 | 0.03 | 0.04 | 0.07 |
| 800 | 0.03 | 0.06 | 0.055 | 0.06 | 0.045 | 0.07 |
| 1000 | 0.025 | 0.05 | 0.045 | 0.035 | 0.045 | 0.065 |

Table 6: Goodness-of-fit level experiment using a Gaussian-Bernoulli Restricted Boltzmann Machine varying the dimension.

| Dimension | FSSD | LSD | KSDAggInc $R = 1$ | KSDAggInc $R = 100$ | KSDAggInc $R = 200$ | KSDAggCom |
|---|---|---|---|---|---|---|
| 20 | 0.02 | 0.055 | 0.045 | 0.06 | 0.065 | 0.05 |
| 40 | 0.04 | 0.055 | 0.07 | 0.055 | 0.065 | 0.07 |
| 60 | 0.04 | 0.055 | 0.06 | 0.04 | 0.05 | 0.06 |
| 80 | 0.015 | 0.04 | 0.045 | 0.04 | 0.035 | 0.05 |

## D.4 Aggregation experiments

We illustrate in Figure 4 the benefits of the aggregation procedure by starting from a 'collection' consisting of only the median bandwidth and increasing the size of the collection by adding more bandwidths. In all three settings, we observe that the power for the test with only the median bandwidth is low. As we increase the number of bandwidths, the power first increases as the test has access to 'better-suited' bandwidths.

For MMDAggInc and KSDAggInc, once the optimal bandwidth is included in the collection, the power reaches a plateau. We do not pay a price in power for considering more bandwidths (or kernels), and so the user is encouraged to consider many kernels with various bandwidths. For the unscaled Gaussian kernel, we are essentially aggregating over kernel matrices which interpolate between the identity matrix (as the bandwidth goes to 0) and the matrix of ones (as the bandwidth goes to $\infty$).

The HSICAggInc case is more challenging: since there are pairs of kernels, the total number of bandwidth combinations grows rapidly (*e.g.* $b$ bandwidths for each kernel corresponds to $b^2$ pairs of kernels). In this case, we observe a significant decay in test power once more than 49 bandwidths are considered.

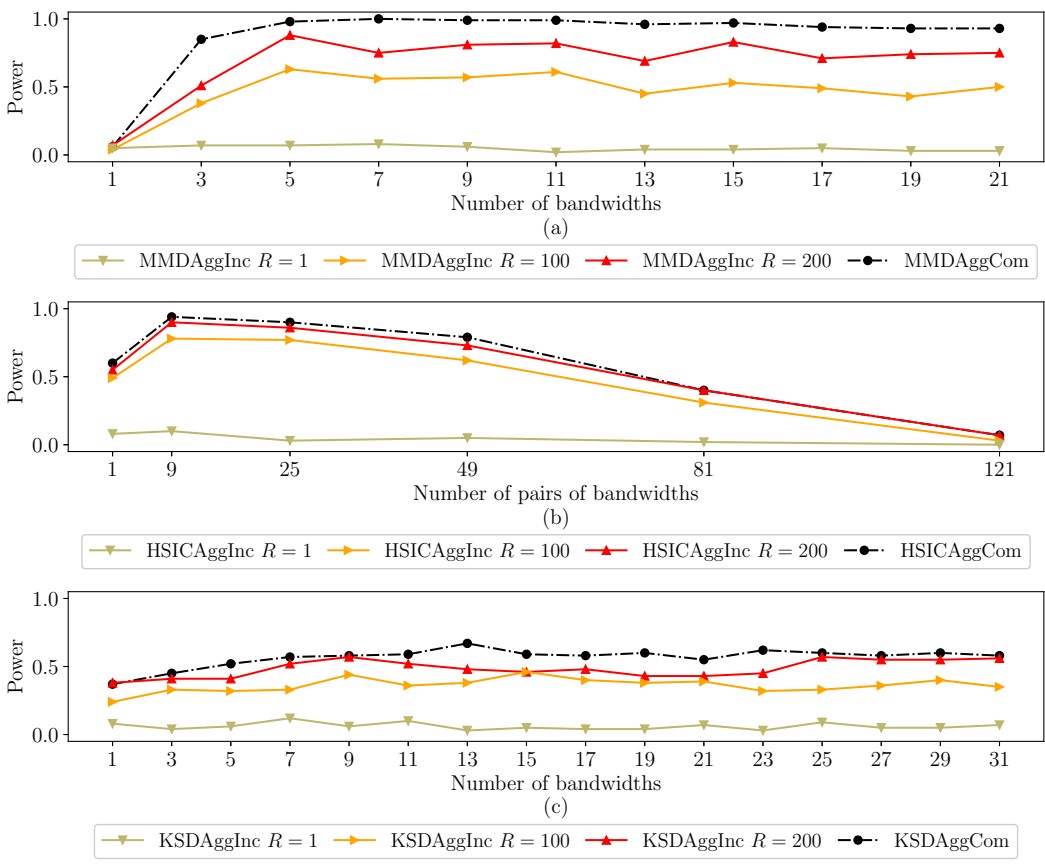

Figure 4: Increasing the collection of bandwidths in the experimental setting of Figure 1 for the *(a)* two-sample , *(b)* independence, and *(c)* goodness-of-fit problems. The power results are averaged over 100 repetitions.

## D.5 Parameter choice for L1 IMQ and Cauchy RFF tests of Huggins and Mackey (2018)

As in the experiments section of Huggins and Mackey (2018) (and as for FSSD), 10 features are used when running Cauchy RFF and L1 IMQ. In their implementation for their experiments, they draw 5000 samples from the unnormalized density for covariance matrix estimation to simulate the null

hypothesis (code: `RFDH0SimCovDrawV(n_draw=5000)`). This procedure causes long runtimes of roughly 16 seconds; this is much more computationally expensive than simulating the null using a wild bootstrap as KSDAggInc does.

We tried different values for `n_draw` and found that using `n_draw` $= 500$ has almost no effect on the test power and reduces the runtimes from 16 seconds for `n_draw` $= 5000$, to 2 seconds (as reported in Figure 1 (l)) for `n_draw` $= 500$. We tried smaller values than $500$ for `n_draw` but this drastically decreased test power. We also verified that the test still has well-calibrated level when using `n_draw` $= 500$. We have used this tuned parameter `n_draw` $= 500$ in our experiments in Figure 1 (i–l). In Figure 5, we show the power and runtime differences when using 500 or 5000 for `n_draw` for L1 IMQ and Cauchy RFF in the setting considered in Figure 1 (i–l), we also plot the test power and runtimes achieved by KSDAggInc $R = 200$.

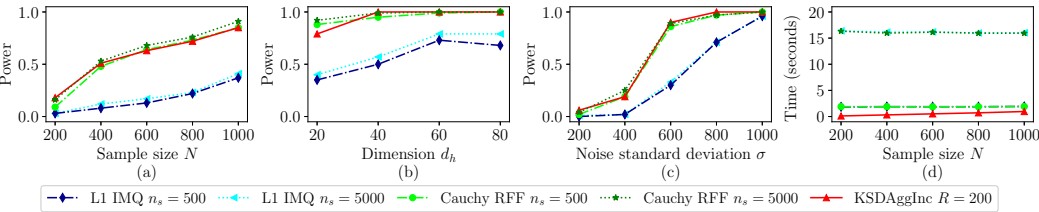

Figure 5: Goodness-of-fit experiment using a Gaussian-Bernoulli Restricted Boltzmann Machine. We consider L1 IMQ and Cauchy RFF tests of Huggins and Mackey (2018) drawing either $n_s =$ `n_draw` $= 5000$ or $n_s =$ `n_draw` $= 500$ samples to simulate the null hypothesis. The power results are averaged over 100 repetitions and the runtimes over 20 repetitions.

# E  Discussions

In this section, we provide detailed discussions on several subjects. We present the motivation behind the definitions of the MMD and HSIC estimators of Equations (8) and (9). We also explain how to define a different incomplete MMD $U$-statistic which is better-suited to the case of unbalanced sample sizes, we point out the challenges arising from working with this estimator. Finally, we provide details on comparison with related work, and on future research directions.

## E.1  Motivation behind expressions (8) and (9)

For the two-sample problem, Equation (26) of Kim et al. (2022, Section 6.1) gives an expression of the MMD $U$-statistic as

$$U_1 = \frac{1}{\left|\mathbf{i}_2^m\right|\left|\mathbf{i}_2^n\right|} \sum_{(i,i')\in\mathbf{i}_2^m} \sum_{(j,j')\in\mathbf{i}_2^n} h_k^{\mathrm{MMD}}(X_i, X_{i'}; Y_j, Y_{j'}).$$

Now, one way to construct an incomplete MMD $U$-statistic would be to replace those two complete sums above with two incomplete sums (see Appendix E.2), but we do not want to take this approach in order to keep a unified framework across the three testing frameworks. We instead take the summation over $(i, i') \in \mathbf{i}_2^N$ and obtain an estimator

$$U_2 = \frac{1}{\left|\mathbf{i}_2^N\right|} \sum_{(i,i')\in\mathbf{i}_2^N} h_k^{\mathrm{MMD}}(X_i, X_{i'}; Y_i, Y_{i'}),$$

where $N = \min(m, n)$. By assuming $N = m$, we denote by $\{L_1, \ldots, L_n\}$ a random permutation of $\{1, \ldots, n\}$. As noted by Kim et al. (2022, Section 6.1), the expectation of

$$\frac{1}{\left|\mathbf{i}_2^N\right|} \sum_{(i,i')\in\mathbf{i}_2^N} h_k^{\mathrm{MMD}}(X_i, X_{i'}; Y_{L_i}, Y_{L_{i'}})$$

over $\{L_1, \ldots, L_n\}$ is equal to $U_1$. This motivates our choice of incomplete MMD estimator in Equation (8) of our paper, which can be regarded as a generalization of $U_2$ above. Similarly, as

discussed in Kim et al. (2022, Section 6.2), the complete HSIC $U$-statistic in Equation (3) can be viewed as the average of incomplete $U$-statistics. More specifically, let $\{L_1, \ldots, L_{\lfloor N/2 \rfloor}\}$ be a $\lfloor N/2 \rfloor$-tuple uniformly sampled without replacement from $\{1, \ldots, N\}$, and let $\{\tilde{L}_1, \ldots, \tilde{L}_{\lfloor N/2 \rfloor}\}$ be another $\lfloor N/2 \rfloor$-tuple uniformly sampled without replacement from $\{1, \ldots, N\} \setminus \{L_1, \ldots, L_{\lfloor N/2 \rfloor}\}$. Then, the $U$-statistic in Equation (3) is the expectation of

$$\frac{1}{\left|\mathbf{i}_2^{\lfloor N/2 \rfloor}\right|} \sum_{(i_1,i_2) \in \mathbf{i}_2^{\lfloor N/2 \rfloor}} h_{k,\ell}^{\mathrm{HSIC}}(Z_{L_{i_1}}, Z_{L_{i_2}}; Z_{\tilde{L}_{i_1}}, Z_{\tilde{L}_{i_2}}),$$

over $\{L_1, \ldots, L_{\lfloor N/2 \rfloor}, \tilde{L}_1, \ldots, \tilde{L}_{\lfloor N/2 \rfloor}\}$. This motivates the definition of our incomplete HSIC estimator in Equation (9).

## E.2 Incomplete MMD $U$-statistic with unbalanced sample sizes

Our incomplete $U$-statistic of Equation (8) for the two-sample problem is constructed using the minimum between $m$ and $n$. If the sample sizes are of the same order of magnitude, then this is not restrictive since we are interested in using only a subset of entries of the kernel matrix in the first place. However, in the setting in which the difference between $m$ and $n$ is of several orders of magnitude, our estimator in Equation (8) does not effectively incorporate the unbalanced sample sizes. When the sample sizes are highly unbalanced, one could instead consider an alternative incomplete $U$-statistic given as

$$\begin{aligned}
U_{\mathrm{imb}} &= \frac{1}{|\mathcal{D}_m||\mathcal{D}_n|} \sum_{(i,j) \in \mathcal{D}_m} \sum_{(r,s) \in \mathcal{D}_n} h_k^{\mathrm{MMD}}(X_i, X_j; Y_r, Y_s) \\
&= \frac{1}{|\mathcal{D}_m||\mathcal{D}_n|} \sum_{(i,j) \in \mathcal{D}_m} \sum_{(r,s) \in \mathcal{D}_n} \Big( k(X_i, X_j) - k(X_i, Y_s) - k(X_j, Y_r) + k(Y_r, Y_s) \Big).
\end{aligned}$$

This expression, for example, results in a linear-time test for the choices $|\mathcal{D}_m| = c\sqrt{m}$ and $|\mathcal{D}_n| = c'\sqrt{n}$ for positive constants $c$ and $c'$ since $|\mathcal{D}_m||\mathcal{D}_n| = cc'\sqrt{m}\sqrt{n} \leq cc'\max(m,n)$. Other choices of design sizes are also possible to obtain linear-time tests. While this estimator is natural for the unbalanced scenario, the form of the test statistic does not allow us to use a wild bootstrap. Instead, one may need to rely on the permutation procedure to calibrate the test statistic, which leads to several theoretical and practical challenges explained below.

**Theory.** From a theoretical side, it is possible to derive a variance bound (corresponding to Lemma 1) for the alternative estimator $U_{\mathrm{imb}}$. However, deriving a quantile bound (corresponding to Lemma 2) for a permuted version of $U_{\mathrm{imb}}$ is highly non-trivial: the extension of the result of Kim et al. (2022, Theorem 6.3) to the case of the permuted version of $U_{\mathrm{imb}}$ is ongoing work.

**Practice.** Theoretically, the cost of computing $B$ permuted estimates is $\mathcal{O}(B|\mathcal{D}_m||\mathcal{D}_n|)$ which would be the same as if we could use a wild bootstrap. However, in practice, the computational time will be much higher because for each permuted estimate, we need to evaluate the kernel matrix at new permuted pairs (possibly outside of the original design), while for the wild bootstrap we do not need to compute any extra kernel values: this changes the computation times drastically. In order to avoid this, we would need to restrict ourselves to permutations for which we have already computed kernel values using the fact that $h_k^{\mathrm{MMD}}(X_i, Y_s; Y_r, X_j) = -h_k^{\mathrm{MMD}}(X_i, X_j; Y_r, Y_s)$. It remains as future work to study conditions under which the set of such permutations is larger than the set consisting of the identity only, and is also large enough to construct accurate quantiles.

## E.3 Comparison with Li and Yuan (2019)

Li and Yuan (2019) also consider the three testing problems and study minimax optimality/adaptivity of their procedures over Sobolev balls. We now discuss and differentiate the approach by Li and Yuan (2019) from ours. First of all, their tests run in quadratic time and control the probability of type I error only asymptotically, while our proposed tests have well-calibrated non-asymptotic levels over a broader class of null distributions and are computationally efficient. Their theoretical guarantees hold only for the Gaussian kernel and with the smoothness restriction that $s > d/4$ while ours hold for a wide range of kernels (see Equation (13)) and for any $s > 0$ (see Theorem 2). We also point out that while they assume that both densities $p$ and $q$ lie in a Sobolev ball, we only require that their

difference $p - q$ belongs to the Sobolev ball. Note that, they tackle the goodness-of-fit problem in a different way. They do not use the KSD and instead use a one-sample MMD with some expectations of the Gaussian kernel under the model. For a generic model density, one cannot compute such an expectation explicitly and hence cannot use their proposed test. In contrast, the KSD that we consider does not suffer from the same issue and it is more broadly applicable.

## E.4    Potential future research directions

This subsection discusses potential directions for future work.

**Interpretable tests.** When the aggregated test rejects the null hypothesis, the test returns all kernels whose associated single test with adjusted level has rejected the null. We stress that this is done using all of the samples, without resorting to data splitting. Those kernels returned by the test are the ones which are well-suited to detect the difference in densities. They can therefore be analysed and interpreted to obtain some information which can help the user understand how the densities differ from each other. For example, we could observe that the densities differ at some specific lengthscales, from which we can infer whether the distribution shift is local, global, or both. If the kernels use different features, we can get a better understanding of the type of features which capture best the difference in densities. This interpretability of the results of our AggInc tests could be very useful, we will further explore it in upcoming work.

**Beyond linear time tests.** Potential directions for future work include studying the regime with $L \lesssim N$, which corresponds to 'faster than linear' tests. For this sub-linear case, our results do not give a definite answer to the question as to whether the separation rate converges to zero. Future work would focus on either deriving tighter bounds that converge to zero in this regime, or proving that the uniform separation rate is bounded below in this setting.

**Computational and statistical trade-off.** As shown in Theorem 1, the quadratic-time MMD and HSIC tests are minimax rate optimal over Sobolev balls. To the best of our knowledge, it is unknown whether there exists a sub-quadratic time test that achieves the same rate optimality. Indeed, the current literature is mostly silent on optimising the power under computational constraints. Theorem 1 (ii) demonstrates a trade-off between the computational budget and the separation rate focusing on incomplete $U$-statistics, but our result does not tell us whether this trade-off is (universally) tight. We think this is one of the limitations of our work and hope that a follow-up study can make progress on this topic.

**Continuously optimising a kernel without data splitting.** In order to achieve a competitive power performance over a large class of alternatives, we combine finitely many kernels and construct an aggregated test. There is another line of work that considers a continuous collection of kernels (for example, indexed by the bandwidth parameter on the positive line) and chooses the kernel that maximises the (empirical) power. As far as we are aware, the current approach to continuously optimizing a kernel relies on data splitting, which often negatively affects the power performance. While our aggregated tests do not require data splitting and can retain high power even for large collections of kernels, in certain scenarios there might exist some important range of kernels not included in our finite collection which would lead to more powerful tests. It is therefore interesting to further extend our approach to the case of a continuous collection of kernels and investigate its statistical properties. We note that, for MMDAggInc and KSDAggInc, as also observed by Schrab et al. (2021, 2022), we find empirically that increasing the number of bandwidths does not result in a loss of power, and corresponds to using a finer discretisation of the intervals considered. The continuous case is the limit of the discretisation.

**Improving conditions.** Lastly, it would be interesting to see if the polynomial factor of $\beta$ in our power guaranteeing conditions of Theorems 1 and 2 can be improved using a sharper concentration bound. Also, future work can be dedicated to figuring out whether the dependence of $\ln(\ln(N))$ in the adaptive rate can be improved. We leave these important and interesting questions to future work.

# F  Proofs

## F.1  Proof of Proposition 1

The asymptotic level of the goodness-of-fit test using a wild bootstrap follows from the results of Shao (2010), Leucht and Neumann (2013), Chwialkowski et al. (2014, 2016). As pointed out by Schrab et al. (2022) for the complete $U$-statistics, the KSD test statistic and the wild bootstrapped KSD statistics are not exchangeable under the null, and hence non-asymptotic level cannot be proved using the result of Romano and Wolf (2005, Lemma 1).

The non-asymptotic level for the two-sample test follows exactly from the reasoning of Schrab et al. (2021, Propositon 1). The fact that we work with incomplete $U$-statistics rather than with their complete counterparts does not affect the proof of exchangeability of $U_\lambda^1, \ldots, U_\lambda^{B_1+1}$.

For the independence problem, Albert et al. (2022, Proposition 1) prove that the quadratic-time HSIC estimator and the permuted test statistics are exchangeable under the null hypothesis, it remains to be shown that this also holds in our incomplete setting using a wild bootstrap. Assuming that exchangeability under the null holds, the desired non-asymptotic level $\alpha$ can then be guaranteed using the result of Romano and Wolf (2005, Lemma 1), exactly as done by Albert et al. (2022, Proposition 1).

We now prove that $U_\lambda^1, \ldots, U_\lambda^{B_1+1}$ for the independence problem are exchangeable under the null. Since $U_\lambda^1, \ldots, U_\lambda^{B_1}$ are i.i.d. given the data, they are exchangeable under the null. So, we need to prove that

$$\sum_{(i,j)\in\mathcal{D}_{\lfloor N/2\rfloor}} h_{k,\ell}^{\mathrm{HSIC}}\left(Z_i, Z_j, Z_{i+\lfloor N/2\rfloor}, Z_{j+\lfloor N/2\rfloor}\right) \tag{19}$$

is, under the null, distributed like

$$\sum_{(i,j)\in\mathcal{D}_{\lfloor N/2\rfloor}} \epsilon_i\epsilon_j h_{k,\ell}^{\mathrm{HSIC}}\left(Z_i, Z_j, Z_{i+\lfloor N/2\rfloor}, Z_{j+\lfloor N/2\rfloor}\right) \tag{20}$$

where $\epsilon_1, \ldots, \epsilon_N$ are i.i.d. Rademacher random variables. Using the result of Schrab et al. (2021, Appendix B, Proposition 11), considering the identity $s_1(i) = i$ for $i = 1, \ldots, 2\lfloor N/2\rfloor$, and the swap function $s_{-1}(i) = i + \lfloor N/2\rfloor$ and $s_{-1}(i + \lfloor N/2\rfloor) = i$ for $i = 1, \ldots, 2\lfloor N/2\rfloor$, we have

$$\sum_{(i,j)\in\mathcal{D}_{\lfloor N/2\rfloor}} \epsilon_i\epsilon_j h_{k,\ell}^{\mathrm{HSIC}}\left(Z_i, Z_j, Z_{i+\lfloor N/2\rfloor}, Z_{j+\lfloor N/2\rfloor}\right)$$

$$= \frac{1}{4}\sum_{(i,j)\in\mathcal{D}_{\lfloor N/2\rfloor}} h_k^{\mathrm{MMD}}\left(X_i, X_j; X_{i+\lfloor N/2\rfloor}, X_{j+\lfloor N/2\rfloor}\right)\left(\epsilon_i\epsilon_j h_\ell^{\mathrm{MMD}}\left(Y_i, Y_j; Y_{i+\lfloor N/2\rfloor}, Y_{j+\lfloor N/2\rfloor}\right)\right)$$

$$= \frac{1}{4}\sum_{\substack{(i,j)\in\\ \mathcal{D}_{\lfloor N/2\rfloor}}} h_k^{\mathrm{MMD}}\left(X_i, X_j; X_{i+\lfloor N/2\rfloor}, X_{j+\lfloor N/2\rfloor}\right)h_\ell^{\mathrm{MMD}}\left(Y_{s_{\epsilon_i}(i)}, Y_{s_{\epsilon_j}(j)}; Y_{s_{\epsilon_i}(i+\lfloor N/2\rfloor)}, Y_{s_{\epsilon_j}(j+\lfloor N/2\rfloor)}\right)$$

$$= \left|\mathcal{D}_{\lfloor N/2\rfloor}\right|\overline{\mathrm{HSIC}}_{k,\ell}\left(\mathbb{Z}_N^\epsilon; \mathcal{D}_{\lfloor N/2\rfloor}\right)$$

where $\mathbb{Z}_N^\epsilon := \left(\left(X_i, Y_{s_{\epsilon_i}(i)}\right)\right)_{1\le i\le N}$ with $Y_{s_{\epsilon_i}(i)} \in \{Y_i, Y_{i+\lfloor N/2\rfloor}\}$ for $i = 1, \ldots, \lfloor N/2\rfloor$ and $Y_{s_{\epsilon_i}(i)} \in \{Y_i, Y_{i-\lfloor N/2\rfloor}\}$ for $i = \lfloor N/2\rfloor + 1, \ldots, 2\lfloor N/2\rfloor$. Now, under the null, the variables $(X_i)_{1\le i\le N}$ and $(Y_i)_{1\le i\le N}$ are independent, so $\mathbb{Z}_N^\epsilon$ is distributed like $\mathbb{Z}_N$. We deduce that $\overline{\mathrm{HSIC}}_{k,\ell}\left(\mathbb{Z}_N; \mathcal{D}_{\lfloor N/2\rfloor}\right)$ and $\overline{\mathrm{HSIC}}_{k,\ell}\left(\mathbb{Z}_N^\epsilon; \mathcal{D}_{\lfloor N/2\rfloor}\right)$ have the same distribution under the null, and hence, that the terms in Equations (19) and (20) also have the same distribution under the null. We deduce that $U_\lambda^1, \ldots, U_\lambda^{B_1+1}$ for the independence problem are exchangeable under the null, which completes the proof.

## F.2  Proof of Lemma 1

Consider the case of fixed design. Using the variance expression of Lee (1990, Theorem 2, p. 190), we have

$$\mathrm{var}(\overline{U}) = \frac{f_1\sigma_1^2 + f_2\sigma_2^2}{|\mathcal{D}|^2}$$

where $f_i$ is the number of pairs of sets in the design $\mathcal{D}$ that have $i$ elements in common. The pairs of sets in $\mathcal{D}$ with 2 elements in common are $\{(\{i,j\},\{i,j\}) : (i,j) \in \mathcal{D}\}$, so $f_2 = |\mathcal{D}|$. We now calculate the number of pairs of sets in $\mathcal{D}$ with 1 element in common. We start with a pair $(i,j) \in \mathcal{D}$ (there are $|\mathcal{D}|$ such pairs). The number of pairs in $\mathcal{D}$ which have one element in common with $(i,j)$ is upper bounded by the number of pairs in $\mathbf{i}_2^N$ which have one element in common with $(i,j)$, those are $\{\{i,r\} : 1 \le r \le N, r \neq i\} \cup \{\{j,r\} : 1 \le r \le N, r \neq j\}$ of size smaller than $2N$. We deduce that $f_1 \le 2N|\mathcal{D}|$. Combining those results, we obtain

$$\mathrm{var}(\overline{U}) \le \frac{f_1 \sigma_1^2 + f_2 \sigma_2^2}{|\mathcal{D}|^2} \le \frac{2N}{|\mathcal{D}|}\sigma_1^2 + \frac{1}{|\mathcal{D}|}\sigma_2^2$$

as desired.

Let us now consider the random design case. Recall that using the variance expression of the complete $U$-statistic of Lee (1990, Theorem 3 p. 12), which we denote $U$, we can obtain that

$$\mathrm{var}(U) \lesssim \frac{\sigma_1^2}{N} + \frac{\sigma_2^2}{N^2}$$

as done by Kim et al. (2022, Appendix E) and Albert et al. (2022, Lemma 10). Using the result of Lee (1990, Theorem 4 p. 193), the variance of the incomplete $U$-statistic $\overline{U}$ can be expressed in terms of the variance of the complete $U$-statistic $U$. For random design with replacement, we have

$$\begin{aligned} \mathrm{var}(\overline{U}) &= \frac{\sigma_2^2}{|\mathcal{D}|} + \left(1 - \frac{1}{|\mathcal{D}|}\right)\mathrm{var}(U) \\ &\lesssim \frac{\sigma_1^2}{N} + \left(\frac{1}{|\mathcal{D}|} + \frac{1}{N^2}\right)\sigma_2^2 \\ &\lesssim \frac{\sigma_1^2}{N} + \frac{1}{|\mathcal{D}|}\sigma_2^2. \end{aligned}$$

Letting $S := N(N-1)/2$, for random design without replacement, we have

$$\begin{aligned} \mathrm{var}(\overline{U}) &= \frac{S - |\mathcal{D}|}{|\mathcal{D}|(S-1)}\sigma_2^2 + \frac{S}{S-1}\left(1 - \frac{1}{|\mathcal{D}|}\right)\mathrm{var}(U) \\ &\lesssim \frac{\sigma_1^2}{N} + \left(\frac{1}{|\mathcal{D}|} + \frac{1}{N^2}\right)\sigma_2^2 \\ &\lesssim \frac{\sigma_1^2}{N} + \frac{1}{|\mathcal{D}|}\sigma_2^2. \end{aligned}$$

We have used the fact that $\frac{1}{N^2} \le \frac{1}{|\mathcal{D}|}$ since $|\mathcal{D}| \le N^2$.

### F.3 Proof of Lemma 2

We rely on the concentration bound for i.i.d. Rademacher chaos of de la Peña and Giné (1999, Corollary 3.2.6) which as presented in Kim et al. (2022, Equation (39)) takes the form

$$\mathbb{P}_\epsilon\left(\left|\sum_{(i,j)\in\mathbf{i}_2^N}\epsilon_i\epsilon_j a_{i,j}\right| \ge t\right) \le 2\exp\left(-Ct\left(\sum_{(i,j)\in\mathbf{i}_2^N} a_{i,j}^2\right)^{-1}\right)$$

for some constant $C > 0$ and for every $t \ge 0$, where $\epsilon_1, \ldots, \epsilon_N$ are i.i.d. Rademacher random variables taking values in $\{-1, 1\}$. Letting

$$a_{i,j} := \frac{h(Z_i, Z_j)}{|\mathcal{D}|}\mathbf{1}\Big[(i,j) \in \mathcal{D}\Big] \qquad \text{for} \qquad (i,j) \in \mathbf{i}_2^N,$$

where $\mathbf{1}$ denotes the indicator function, we obtain

$$\mathbb{P}_\epsilon\left(\frac{1}{|\mathcal{D}|}\left|\sum_{(i,j)\in\mathcal{D}}\epsilon_i\epsilon_j h(Z_i,Z_j)\right|\geq t \;\Big|\; \mathbb{Z}_N,\mathcal{D}\right) \leq 2\exp\left(-Ct\left(\frac{1}{|\mathcal{D}|^2}\sum_{(i,j)\in\mathcal{D}}h(Z_i,Z_j)^2\right)^{-1}\right)$$

$$\leq 2\exp\left(-Ct\left(\frac{1}{|\mathcal{D}|^2}\sum_{(i,j)\in\mathbf{i}_2^N}h(Z_i,Z_j)^2\right)^{-1}\right)$$

which concludes the proof.

### F.4 Proof of Theorem 1

We start by reviewing the steps of the proofs of Albert et al. (2022) and Schrab et al. (2021, 2022) who prove that, for each of the three respective testing frameworks, a sufficient condition to ensure control of the probability of type II error for the quadratic-time test is the existence of a constant $C > 0$ such that

$$\|p-q\|_2^2 \geq \|(p-q)-T_\lambda(p-q)\|_2^2 + C\frac{1}{N}\frac{\ln(1/\alpha)}{\beta}\sigma_{2,\lambda}. \tag{21}$$

Those quadratic-time tests use the complete $U$-statistics defined in Equations (1), (3) and (5), which we denote as $U_\lambda$. The key results for their proofs rely on deriving variance and quantile bounds.

The variance bound is of the form

$$\mathrm{var}(U_\lambda) \lesssim \frac{1}{N}\sigma_1^2 + \frac{1}{N^2}\sigma_2^2 \tag{22}$$

where they show that, for $h_\lambda \in \{h_{k_\lambda}^{\mathrm{MMD}}, h_{k_\lambda,\ell_\mu}^{\mathrm{HSIC}}, h_{k_\lambda,p}^{\mathrm{KSD}}\}$ defined in Equations (2), (4) and (6), we have

$$\sigma_{1,\lambda}^2 := \mathrm{var}\big(\mathbb{E}\big[h_\lambda(Z,Z')\big|Z'\big]\big) \lesssim \big\|T_\lambda(p-q)\big\|_2^2$$

and

$$\sigma_{2,\lambda}^2 := \mathrm{var}(h_\lambda(Z,Z')) = \mathbb{E}\big[h_\lambda(Z,Z')^2\big] \lesssim \frac{1}{\lambda_1\cdots\lambda_d} \tag{23}$$

where the last inequality holds only for $h_\lambda \in \{h_{k_\lambda}^{\mathrm{MMD}}, h_{k_\lambda,\ell_\mu}^{\mathrm{HSIC}}\}$.

The quantile bound (Schrab et al., 2021, Proposition 4) is of the form

$$\mathbb{P}\left(\widehat{q}_{1-\alpha}^{\lambda,U,B_1} \lesssim \frac{1}{N}\frac{1}{\sqrt{\delta}}\ln\left(\frac{1}{\alpha}\right)\sigma_{2,\lambda}\right) \geq 1-\delta$$

for $\delta \in (0,1)$, where $\widehat{q}_{1-\alpha}^{\lambda,U,B_1}$ is the quantile obtained using $B_1$ wild bootstrapped similarly to the one defined in Equation (14) but using the complete $U$-statistic. Relying on Dvoretzky–Kiefer–Wolfowitz inequality (Dvoretzky et al., 1956; Massart, 1990), Schrab et al. (2021, Proposition 4) show that it suffices to prove the bound

$$\mathbb{P}\left(\widehat{q}_{1-\alpha}^{\lambda,U,\infty} \lesssim \frac{1}{N}\frac{1}{\sqrt{\delta}}\ln\left(\frac{1}{\alpha}\right)\sigma_{2,\lambda}\right) \geq 1-\delta \tag{24}$$

for the true wild bootstrap quantile $\widehat{q}_{1-\alpha}^{\lambda,U,\infty}$ without finite approximation.

Combining those variance and quantile bounds using Chebyshev's inequality (Chebyshev, 1899), they obtain a condition guaranteeing power in terms of the MMD, HSIC and KSD. By expressing these three measures as an RHKS inner product

$$\langle p-q, T_\lambda(p-q)\rangle = \frac{1}{2}\Big(\|p-q\|_2^2 + \|T_\lambda(p-q)\|_2^2 - \|(p-q)-T_\lambda(p-q)\|_2^2\Big),$$

they obtain the condition in Equation (21) which guarantees high power in terms of $\|p-q\|_2^2$. Albert et al. (2022) and Schrab et al. (2021) then derive the minimax rate $N^{-2s/(4s+d)}$ over the Sobolev

ball $\mathcal{S}_d^s(R)$ for the independence and two-sample tests using the bandwidths $\lambda_i^* := N^{-2/(4s+d)}$ for $i = 1, \ldots, d$.

For Theorem 1, we need to obtain the condition in Equation (21) with $N$ replaced by $L/N$. Hence, following their reasoning, in order to prove Theorem 1 (i) & (ii), it suffices to derive variance and quantiles bounds for incomplete $U$-statistics which have the form of Equations (22) and (24) with $N$ replaced by $L/N$, which we now do.

Using the variance bound for incomplete $U$-statistics $\overline{U}_\lambda$ of Lemma 1, together with the fact that the design size $L := |\mathcal{D}|$ is smaller than $N^2$ so that $1/L = L/L^2 \leq N^2/L^2$, we obtain for fixed design that

$$\operatorname{var}(\overline{U}_\lambda) \lesssim \frac{N}{L}\sigma_{1,\lambda}^2 + \frac{1}{L}\sigma_{2,\lambda}^2 \lesssim \frac{N}{L}\sigma_{1,\lambda}^2 + \left(\frac{N}{L}\right)^2 \sigma_{2,\lambda}^2.$$

We get the same bound for random design since

$$\operatorname{var}(\overline{U}_\lambda) \lesssim \frac{1}{N}\sigma_{1,\lambda}^2 + \frac{1}{L}\sigma_{2,\lambda}^2 \lesssim \frac{N}{L}\sigma_{1,\lambda}^2 + \left(\frac{N}{L}\right)^2 \sigma_{2,\lambda}^2,$$

as desired.

For the quantile bound, we use Lemma 2 which, for $A_\lambda^2 := L^{-2}\sum_{(i,j)\in\mathbf{i}_2^N} h_\lambda(Z_i, Z_j)^2$, gives that there exists some[9] $C > 0$ such that

$$\mathbb{P}_\epsilon\left(\overline{U}_\lambda^\epsilon \geq t \,\middle|\, \mathbb{Z}_N, \mathcal{D}\right) \leq \mathbb{P}_\epsilon\left(\left|\overline{U}_\lambda^\epsilon\right| \geq t \,\middle|\, \mathbb{Z}_N, \mathcal{D}\right) \leq 2\exp\left(-\frac{t}{CA_\lambda}\right).$$

Setting $\alpha := 2\exp(-t/CA_\lambda)$, we obtain

$$\widehat{q}_{1-\alpha}^{\lambda,\overline{U},\infty} = t = CA_\lambda \ln(2/\alpha).$$

For $\delta \in (0,1)$, using Markov's inequality, we obtain

$$\mathbb{P}\left(A_\lambda^2 \leq \frac{1}{\delta}\mathbb{E}[A_\lambda^2]\right) \geq 1 - \delta$$

where

$$\mathbb{E}[A_\lambda^2] = \mathbb{E}\left[\frac{1}{L^2}\sum_{(i,j)\in\mathbf{i}_2^N} h(Z_i, Z_j)^2\right] = \frac{N(N-1)}{L^2}\mathbb{E}[h_\lambda(Z, Z')^2] \lesssim \frac{N^2}{L^2}\sigma_{2,\lambda}^2$$

using Equation (23). We deduce that

$$1 - \delta \leq \mathbb{P}\left(A_\lambda^2 \leq \frac{1}{\delta}\mathbb{E}[A_\lambda^2]\right)$$

$$= \mathbb{P}\left(\widehat{q}_{1-\alpha}^{\lambda,\overline{U},\infty} \leq C\frac{1}{\sqrt{\delta}}\ln\left(\frac{2}{\alpha}\right)\sqrt{\mathbb{E}[A_\lambda^2]}\right)$$

$$\leq \mathbb{P}\left(\widehat{q}_{1-\alpha}^{\lambda,\overline{U},\infty} \lesssim \frac{1}{\sqrt{\delta}}\frac{N}{L}\ln\left(\frac{1}{\alpha}\right)\sigma_{2,\lambda}\right)$$

where we absorbed the constant $C$ in the notation '$\lesssim$', and where we used the fact that $\ln(2/\alpha) \lesssim \ln(1/\alpha)$ since $\alpha \in (0, e^{-1})$. This concludes the proof.

### F.5 Proof of Theorem 3

#### F.5.1 Proof of Theorem 3 (i)

In this setting, we consider as proposed by Kim et al. (2022, Equation 32) the permuted HSIC complete $U$-statistic

$$U_N^\pi := \frac{1}{|\mathbf{i}_4^N|}\sum_{(i,j,r,s)\in\mathbf{i}_4^N} h_{k,\ell}^{\mathrm{HSIC}}\left((X_i, Y_{\pi_i}), (X_j, Y_{\pi_j}), (X_r, Y_{\pi_r}), (X_s, Y_{\pi_s})\right)$$

---

[9]For simplicity of notation, we work with $C^{-1} > 0$ rather than with $C > 0$.

for a permutation $\pi$ of the indices $\{1, \ldots, N\}$, and for $h_{k,\ell}^{\mathrm{HSIC}}$ as defined in Equation (4).

Applying the exponential concentration bound of Kim et al. (2022, Theorem 6.3), which uses the result of de la Peña and Giné (1999, Theorem 4.1.12), we obtain that there exist constants $C_1, C_2 > 0$ such that

$$\mathbb{P}_\pi(U_N^\pi \geq t \mid \mathbb{Z}_N) \leq C_1 \exp\left(-C_2 \min\left(\frac{Nt}{\Lambda_N}, \frac{Nt^{2/3}}{M_N^{2/3}}\right)\right) \tag{25}$$

where

$$\Lambda_N^2 := \frac{1}{N^4} \sum_{i=1}^N \sum_{j=1}^N \sum_{r=1}^N \sum_{s=1}^N k_\lambda(X_i, X_j)^2 \ell_\mu(Y_r, Y_s)^2$$

and

$$
\begin{aligned}
M_N &:= \max_{1 \leq i,j,r,s \leq N} \left| k_\lambda(X_i, X_j) \ell_\mu(Y_r, Y_s) \right| \\
&= \max_{1 \leq i,j,r,s \leq N} \left| \prod_{a=1}^{d_x} \frac{1}{\lambda_a} K_a\left(\frac{(X_i)_a - (X_j)_a}{\lambda_a}\right) \prod_{b=1}^{d_y} \frac{1}{\lambda_b} L_b\left(\frac{(Y_r)_b - (Y_s)_b}{\lambda_b}\right) \right| \\
&\lesssim \frac{1}{\lambda_1 \ldots \lambda_{d_x} \mu_1 \ldots \mu_{d_y}} \\
&= \frac{1}{\lambda_1 \cdots \lambda_d}
\end{aligned}
$$

since the functions $K_1, \ldots, K_{d_x}$ and $L_1, \ldots, L_{d_y}$ are bounded, and where we recall our notational convention that $d := d_x + d_y$ and $\lambda_{d_x+i} := \mu_i$ for $i = 1, \ldots, d_y$.

Using the reasoning of Schrab et al. (2021, Proposition 3), we see that the results of Albert et al. (2022, Equations C.17, C.18 & C.19) hold not only for the Gaussian kernel but more generally for any kernels of the form of Equation (13). Those results give us that

$$
\begin{aligned}
\mathbb{E}\left[k_\lambda(X_1, X_2)^2 \ell_\mu(Y_1, Y_2)^2\right] &\lesssim \frac{1}{\lambda_1 \ldots \lambda_{d_x} \mu_1 \ldots \mu_{d_y}} = \frac{1}{\lambda_1 \cdots \lambda_d}, \\
\mathbb{E}\left[k_\lambda(X_1, X_2)^2 \ell_\mu(Y_1, Y_3)^2\right] &\lesssim \frac{1}{\lambda_1 \cdots \lambda_d}, \\
\mathbb{E}\left[k_\lambda(X_1, X_2)^2 \ell_\mu(Y_3, Y_4)^2\right] &\lesssim \frac{1}{\lambda_1 \cdots \lambda_d},
\end{aligned}
$$

the constant in the notation '$\lesssim$' depends only on $d$ and $M$, where we recall that by assumption we have $\max\left(\|p_{xy}\|_\infty, \|p_x\|_\infty, \|p_y\|_\infty\right) \leq M$. We deduce that

$$\mathbb{E}\left[\Lambda_N^2\right] \lesssim \frac{1}{\lambda_1 \cdots \lambda_d}.$$

As explained in Appendix F.4, by relying on Dvoretzky–Kiefer–Wolfowitz inequality (Dvoretzky et al., 1956; Massart, 1990) as done by Schrab et al. (2021, Proposition 4), it is sufficient to prove upper bounds for the true permutation quantile $\widehat{q}_{1-\alpha}^{\lambda, \infty}$ without finite approximation. From Equation (25), we obtain that this quantile satisfies

$$\widehat{q}_{1-\alpha}^{\lambda, \infty} \lesssim \max\left(\frac{\Lambda_N}{N} \ln\left(\frac{1}{\alpha}\right), \frac{M_N}{N^{3/2}} \ln\left(\frac{1}{\alpha}\right)^{3/2}\right).$$

Using Markov's inequality and bounds obtained above, we get that

$$\widehat{q}_{1-\alpha}^{\lambda, \infty} \lesssim \max\left(\frac{\sqrt{\mathbb{E}\left[\Lambda_N^2\right]}}{\sqrt{\delta} N} \ln\left(\frac{1}{\alpha}\right), \frac{M_N}{N^{3/2}} \ln\left(\frac{1}{\alpha}\right)^{3/2}\right)$$

$$\widehat{q}_{1-\alpha}^{\lambda, \infty} \lesssim \max\left(\frac{\ln(1/\alpha)}{\sqrt{\delta} N \sqrt{\lambda_1 \cdots \lambda_d}}, \frac{\ln(1/\alpha)^{3/2}}{N^{3/2} \lambda_1 \cdots \lambda_d}\right) \tag{26}$$

holds with probability at least $1 - \delta$ where $\delta \in (0, 1)$.

Now, recall that by assumption $4s > d$ and $\lambda_i^* = N^{-2/(4s+d)}$ for $i = 1, \ldots, d$, so that

$$\frac{1}{\lambda_1^* \cdots \lambda_d^*} = N^{2d/(4s+d)} < N \qquad \Longleftrightarrow \qquad N^{-1/2} < \sqrt{\lambda_1^* \cdots \lambda_d^*}$$

which gives

$$\widehat{q}_{1-\alpha}^{\lambda^*, \infty} \lesssim \max\left(\frac{\ln(1/\alpha)}{\sqrt{\delta} N \sqrt{\lambda_1^* \cdots \lambda_d^*}}, \frac{\ln(1/\alpha)^{3/2}}{N \sqrt{\lambda_1^* \cdots \lambda_d^*}}\right)$$

$$\lesssim \frac{\ln(1/\alpha)^{3/2}}{\sqrt{\delta} N \sqrt{\lambda_1^* \cdots \lambda_d^*}}$$

holding with probability at least $1 - \delta$, since $\alpha \in (0, e^{-1})$. By combining this result with the reasoning of Albert et al. (2022) as explained in Appendix F.4, we obtain that the probability of type II error of the test is controlled by $\beta \in (0, 1)$ when

$$\|p - q\|_2^2 \geq \|(p - q) - T_{\lambda^*}(p - q)\|_2^2 + C \frac{1}{N} \frac{\ln(1/\alpha)^{3/2}}{\beta \sqrt{\lambda_1^* \cdots \lambda_d^*}}.$$

We have recovered the correct dependency with respect to $N$ and $\lambda$ with an improved $\alpha$-dependency of $\ln(1/\alpha)^{3/2}$ compared to the $\alpha^{-1/2}$ dependency obtained by Kim et al. (2022, Proposition 8.7). The proof of minimax optimality of the quadratic-time test with fixed bandwidth $\lambda^*$ does not depend on the $\alpha$-dependency and can be derived in both our setting and the one of Kim et al. (2022) using quantiles obtained from permutations by following the reasoning of Albert et al. (2022, Corollary 2). We obtain that the uniform separation rate over the Sobolev ball $\mathcal{S}_d^s(R)$ is, up to a constant, $N^{-2s/(4s+d)}$. The improved $\alpha$-dependency is crucial for deriving the rate of the aggregated quadratic-time test over Sobolev balls because the weights appear in the $\alpha$-term (*i.e.* $\alpha$ is replaced by $\alpha w_\lambda$ which depends on the sample size $N$).

### F.5.2 Proof of Theorem 3 (ii)

Similarly to the proofs of Albert et al. (2022, Corollary 3) and Schrab et al. (2021, Corollary 10), consider

$$\ell^* := \left\lceil \frac{2}{4s+d} \log_2\left(\frac{N}{\ln(\ln(N))}\right)\right\rceil \leq \left\lceil \frac{2}{d} \log_2\left(\frac{N}{\ln(\ln(N))}\right)\right\rceil$$

and the bandwidth $\lambda^* := (2^{-\ell^*}, \ldots, 2^{-\ell^*}) \in \Lambda$ which satisfies

$$\ln\left(\frac{1}{w_{\lambda^*}}\right) \lesssim \ln(\ell^*) \lesssim \ln(\ln(N))$$

as $w_{\lambda^*} := 6\pi^{-2} (\ell^*)^{-2}$, and

$$\frac{1}{2}\left(\frac{N}{\ln(\ln(N))}\right)^{-2/(4s+d)} \leq \lambda_i^* \leq \left(\frac{N}{\ln(\ln(N))}\right)^{-2/(4s+d)}$$

for $i = 1, \ldots, d$. Since $4s > d$, we have

$$\frac{1}{\lambda_1^* \cdots \lambda_d^*} \leq 2^d \left(\frac{N}{\ln(\ln(N))}\right)^{2d/4s+d} \lesssim \frac{N}{\ln(\ln(N))} \qquad \Longleftrightarrow \qquad N^{-1/2} \lesssim \frac{\sqrt{\lambda_1^* \cdots \lambda_d^*}}{\sqrt{\ln(\ln(N))}}.$$

By Equation (26), we get that

$$\widehat{q}_{1-\alpha w_{\lambda^*}}^{\lambda^*, \infty} \lesssim \max\left(\frac{\ln(1/(\alpha w_{\lambda^*}))}{\sqrt{\delta} N \sqrt{\lambda_1^* \cdots \lambda_d^*}}, \frac{\ln(1/(\alpha w_{\lambda^*}))^{3/2}}{N^{3/2} \lambda_1^* \cdots \lambda_d^*}\right) \tag{27}$$

holds with probability at least $1 - \delta$ for $\delta \in (0, 1)$. If the largest term in Equation (27) is the first one, then we get

$$\widehat{q}_{1-\alpha w_{\lambda^*}}^{\lambda^*, \infty} \lesssim \frac{\ln(1/\alpha) + \ln(1/w_{\lambda^*}))}{\sqrt{\delta} N \sqrt{\lambda_1^* \cdots \lambda_d^*}},$$

$$\widehat{q}_{1-\alpha w_{\lambda^*}}^{\lambda^*, \infty} \lesssim \frac{\ln(\ln(N))}{\sqrt{\delta} N \sqrt{\lambda_1^* \cdots \lambda_d^*}}, \tag{28}$$

where the constant in '$\lesssim$' also depends on $\alpha$. The result then follows exactly as in the proofs of Albert et al. (2022, Corollary 3) and Schrab et al. (2021, Corollary 10). So, we consider the case where the second term in Equation (27) is the largest one, so that

$$
\begin{aligned}
\widehat{q}_{1-\alpha w_{\lambda^*}}^{\lambda^*,\infty} &\lesssim \frac{\ln\big(1/(\alpha w_{\lambda^*})\big)^{3/2}}{N^{3/2}\lambda_1^*\cdots\lambda_d^*} \\
&\lesssim \frac{\ln\big(1/w_{\lambda^*}\big)^{3/2}}{N\lambda_1^*\cdots\lambda_d^*} N^{-1/2} \\
&\lesssim \frac{\ln(\ln(N))^{3/2}}{N\lambda_1^*\cdots\lambda_d^*} \frac{\sqrt{\lambda_1^*\cdots\lambda_d^*}}{\sqrt{\ln(\ln(N))}} \\
&= \frac{\ln(\ln(N))}{N\sqrt{\lambda_1^*\cdots\lambda_d^*}}.
\end{aligned}
$$

We have recovered the same dependency as in Equation (28) when considering the first term as the largest one, and the proof then follows exactly the ones of Albert et al. (2022, Corollary 3) and Schrab et al. (2021, Corollary 10). We have treated both cases in Equation (27), we conclude that the uniform separation rate over the Sobolev balls $\big\{\mathcal{S}_d^s(R) : R > 0, s > d/4\big\}$ of the quadratic-time aggregated test using a quantile obtained with permutations is (up to a constant)

$$
\left(\frac{N}{\ln(\ln(N))}\right)^{-2s/(4s+d)}.
$$