# OpenReview forum: "Efficient Aggregated Kernel Tests using Incomplete $U$-statistics"
_NeurIPS.cc/2022/Conference — NeurIPS 2022 Accept_

### Official Review · Reviewer_aMao · 2022-07-01

**Rating:** 6
**Confidence:** 4
**Soundness:** 3 good
**Presentation:** 3 good
**Contribution:** 3 good

**Summary:**

Post rebuttal: after the authors response and reading the other reviews, I updated my score to 6 (from initially 4) see also my comment below.


-------- original review ------

The paper considers a general framework for kernel-based hypothesis testing, where the test statistic is given by a U-statistic. The framework covers Two-sample testing, Independence Testing, and Goodness-of-fit testing.
The paper shows that using an incomplete U-statistic estimate allows to trade-off computational resources for statistical significance, which follows from general theory of incomplete U-statistics.
Furthermore, it adopts recent advances to aggregate such tests over multiple kernels and provides insights into the minimax separation rates over Sobolev balls.
Lastly, the paper provides simple experiments on toy data, illustrating their findings and comparing to some other approaches to tackle the respective testing problems.

**Questions:**

- Are the tests really 'linear-time'? (see comment in limitations)
- l. 152-156: why do you discuss the random design when in the end you are using the deterministic one?
- How should one choose the number of bandwidths ($l$) in practice?

Minor Comments:
- l. 147: define what a degenerate kernel is.
- Type in Equation before line 500: should be $x_{i_1}$,...

**Limitations:**

The work is theoretical and no direct negative societal impact is to be expected.

The theoretical results discuss minimax optimal rates, which leaves the impression that nothing can go wrong. But in practice there remain some parameters that users have to choose, for example how many bandwidths to include in the aggregation. For the experiments the authors only use 4 bandwidths, which in my opinion hardly suffices to illustrate the benefits of this aggregation. On the other hand, it is not clear what happens if too many bandwidths are included.

**Strengths And Weaknesses:**

*Strengths*:
- the work applies to three different testing scenarios and illustrates their close connections.
- Although the use of incomplete U-statistics is not completely new for kernel-based tests (see e.g. Yamada et al ICLR 2019) the provided non-asymptotic tests are relevant and nicely illustrate the trade-off between computational resources and statistical significance.
- The tests provably control type-I error also at finite data, while some of the existing methods, like OST, do not.
- The theoretical insights are concisely stated and the prior results properly attributed. Arguably, though the provided results are rather simple consequences and combinations of prior results.
- The provided Code is clean and it is easy to reproduce the experiments. The code was submitted after the deadline, which might be a violation of the rules!

*Weaknesses*:
- Overall, I think the practical relevance is quite limited:
a) The experiments are limited to very simple toy data sets. While they illustrate the effect of using incomplete U-statistics, the effect of the aggregation procedure is not illustrated at all.
b) Only very few (4!) kernels are aggregated over. IMO this does not suffice to illustrate the benefits of the aggregation procedure.
c) Overall there should be more guidance for practitioners. How many kernels should one choose in practice?
- The authors consider ‘linear-time’ tests (eq. 10), which I think is misleading. By their theoretical results Theorem 1 ii) and considering $L=c N$, the uniform separation rate is not guaranteed to converge to zero. I thus also think that the gray box on page 7 is actually wrong. IMO it should be changed to the following: let $L=N^{1+a}$. Then for $a=1$ one recovers the minimax rate. For $a> 0$ the rate still converges to zero, but slower. For $a\leq 0$ there is no guarantee. **Overall the provided theory does not provide guarantees for linear-time tests**.
- For the two-sample problem, the provided tests cannot handle imbalanced samples in a meaningful way. The provided approach simply truncates data ($N=min(m,n)$ in line 131).
- I think the presentation of the initial statistics (1) and (3) is suboptimal. For unexperienced readers it seems that these statistics scale like N^4. So I think it would be better to directly introduce the statistics such that they correspond to the complete U-statistics of (7) and (8). Alternatively, it should be explained why this statistics scale quadratically (no need to tell me in the rebuttal).
- The claim that the “aggregation procedure is known to lead to state-of-the-art powerful tests” (l. 26f) seems a bit biased.
- Prior work (Sutherland (ICLR 2017), Liu (ICML 2020)) showed that continuously optimizing a kernel is quite advantageous and harnesses the beenfits of gradient-based optimization. The present work only allows to combine finitely many (prespecified) kernels.
- The aggregation scheme is a direct adaption from prior work (the authors are transparent about this).

---

> ### Author Response · Authors · 2022-08-02
> **Reply to Reviewer aMao**
>
> We thank reviewer aMao for summarising the strengths of the paper, for suggesting experiments, and for the questions raised.
>
> **Q1:** a) The [...] all.
>
> **A1:** We have now provided additional experiments. In Appendix X, we consider experiments using the real-world MNIST dataset (dimension 784) and observe the same trends as on the toy datasets (which satisfy the Sobolev smoothness assumption). We have also added an experiment which illustrates the benefits of the aggregation procedure. See the main reply common to all reviewers for details.
>
> **Q2:** b) Only [...] procedure.
>
> A2: Following your suggestion, we have increased the number of kernels in our experiments. In particular, we now use 21 kernels for MMDAggInc and KSDAggInc, and 25 kernels for HSICAggInc. The new simulation results indicate that the resulting tests still retain high power and still outperform other tests. See the main reply common to all reviewers for details.
>
> **Q3:** c) Overall [...] ?
>
> A3: Our simulation results demonstrate that using 20-25 kernels seems to present competitive performance under the considered settings. We therefore recommend 20-25 kernels to use in practice, while it is certainly possible to find a better option under different scenarios. We will make this point clear in the final version. See also the main reply common to all reviewers for details.
>
>
> **Q4:** The authors [...] tests.
>
> **A4:** In this paper, we propose efficient tests whose computational cost $L$ (as a function of the sample size $N$) can be chosen by the user. We study the theoretical properties of such tests. In particular, we obtain that if the test is quadratic (i.e. $L\asymp N^2$) then it is minimax optimal. If the test is between linear and quadratic (i.e. $N\lesssim L \lesssim N^2$), we show that there is a price to pay in the minimax rate for computational efficiency. If the test is linear or faster (i.e. $L \lesssim N$), our theoretical results do not guarantee that the rate converges to zero. We believe the gray box on page 7 is correct given that we think of the symbols $<$ and $\leq$ as 'up to a constant', this is we care about $L$ only as a function of $N$. The notation being confusing, we propose to replace it by $\lesssim$ as done in this discussion and emphasizing explicitly in the main text that this means 'up to a constant'.
> We did not write $L$ as $N^{1+a}$ in order to be general, our statements always for cases such as $L\asymp N \log(N)$.
>
> We agree that our theoretical results do not provide guarantees for linear-time tests, the results quantify how the upper bound decays between the minimax rate for quadratic time and a constant for linear time. For anything faster than linear, say $L=N\log(\log(N))$, the rate is guaranteed to converge to 0, in this case at a slow rate of $\log(\log(N))^{-2s/(4s+d)}$. In the experiments, we consider linear tests with $L=cN$ for fixed values of $c$ in order to compare the test performance against other linear-time tests.
>
> **Q5:** For [...] line 131).
>
> **A5:** The reason this truncation does not impact our theoretical results is because of the assumptions that the sample sizes are balanced so that $m\asymp n$ (line 84). For the incomplete $U$-statistic we essentially choose which pairs of data points to consider in the kernel matrix, for computational efficiency we do not consider all the pairs. By truncating the data ($N = \text{min}(n,m)$), we are essentially restricting the number of pairs to choose from but we are still choosing the same number. So, while it is not ideal, we do not think this restriction results in an important loss of power when the sample sizes are balanced. However, we recognise that our test is not well-suited to extreme cases where there are orders of magnitude difference between the sample sizes. Extension to this particular regime is a topic for future work, but would likely require stronger smoothness assumptions on the tested distributions.
>
> **Q6**: I [...] quadratically.
>
> A6: Thanks for your comment. In the final version, we will revise the background section and stress that these can be computed in quadratic time, referring readers to Appendix A for more details.
>
> **Q7**: Prior [...] kernels.
>
> A7: As far as we are aware, the current approach to continuously optimizing a kernel requires data splitting, which negatively affects the power performance. Our aggregated test does not require data splitting and shows a competitive power performance under the considered settings. Nevertheless, we agree that an extension to the case of a continuous collection of kernels (indexed by the bandwidth parameter on the positive real line) is an interesting direction for future work.
>
> **Q8:** l. 152-156
>
> **A8:** We cover both deterministic and random designs for the sake of generality. Indeed, our theory holds for both designs while we focus on the deterministic design in our experiments. We think this general theory will be beneficial for a follow-up study and other related work.

---

> > ### Comment · Reviewer_aMao · 2022-08-06
> > **Update of Review**
> >
> > I thank the authors for their response and providing additional experiments which strengthen the paper.
> >
> > Also, after reading the other reviews, I will upgrade my score from 4 to 6 for now.
> >
> > One last comment regarding "linear" or not. I just want to prevent that the method is sold as "linear", and that's essentially how I read it from the experiments section.
> >
> > Also maybe the authors can think of a better method regarding Q5. Indeed in theory it doesn't matter that one sample is simply truncated. But in practice it would be nice to use those samples (maybe without actually increasing the computational cost).

---

> > > ### Author Response · Authors · 2022-08-09
> > > **Response to Reviewer aMao**
> > >
> > > We warmly thank reviewer aMao for increasing their score!
> > >
> > > We will make sure to clarify the following points in the final version:
> > >
> > > (i) The tests we propose have a computational cost which can be specified by the user (the size of the design between $1$ and $N^2$), there is a tradeoff between test power and computational cost.
> > >
> > > (ii) We provide our theoretical rates in terms of the sample size $N$ working up to a constant. The rate is minimax optimal in the case where the design size grows quadratically with $N$. We quantify exactly how the rate deteriorates from quadratic (minimax) to linear (no guarantee) growth of the design size with respect to the sample size.
> > >
> > > (iii) In practice, one (among many others) possible choice of design size is to use $cN$ for some positive constant $c$. With this choice, the resulting tests are linear-time which allows us to compare them against other linear-time tests in our experiments. However, the assumption for having a rate converging to 0 in (ii) is not satisfied in this setting.
> > >
> > > Regarding the case of imbalanced sample sizes for the two-sample problem, we explain how such an estimator could be defined and point out the challenges that arise from working with it. We will provide such a discussion in the appendix of the final version of the paper.
> > >
> > > Recall that the original quadratic-time MMD estimate is
> > > $$
> > > \frac{1}{|\textbf{i}_2^m| |\textbf{i}_2^n|}
> > > \sum^{(i,j)\in \textbf{i}_2^m}
> > > \sum^{(r,s)\in \textbf{i}_2^n}
> > > h_k^{MMD}(X_i, X_j; Y_r, Y_s)
> > > $$
> > > This is a two-sample complete $U$-statistic and its incomplete version is
> > > $$
> > > \frac{1}{|\mathcal{D}_m| |\mathcal{D}_n|}
> > > \sum^{(i,j)\in \mathcal{D}_m}
> > > \sum^{(r,s)\in \mathcal{D}_n}
> > > h_k^{MMD}(X_i, X_j; Y_r, Y_s)
> > > =
> > > \frac{1}{|\mathcal{D}_m| |\mathcal{D}_n|}
> > > \sum^{(i,j)\in \mathcal{D}_m}
> > > \sum^{(r,s)\in \mathcal{D}_n}
> > > \Big(k(X_i,X_j)  - k(X_i,Y_s) - k(X_j,Y_r) + k(Y_r,Y_s)\Big).
> > > $$
> > > This expression, for example, result in a linear-time test for the choice $|\mathcal{D}_m| = c \sqrt{m}$ and $|\mathcal{D}_n| = c' \sqrt{n}$ for positive constants $c$ and $c'$ since $|\mathcal{D}_m| |\mathcal{D}_n| = c c' \sqrt{m} \sqrt{n} \leq c c' \text{max}(m,n)$. Other choices of design sizes are possible to obtain linear-time tests.
> > >
> > > It is worth pointing out, however, that a wild bootstrap cannot be used with such an estimator. In order to calibrate the test non-asymptotically, permutations should be used instead. We now describe several challenges associated with the permutation approach.
> > >
> > > **Theory:** We believe we can easily obtain a variance bound equivalent to Lemma 1 which holds for this estimate. However, we believe that deriving a quantile bound (equivalent of Lemma 2) for permuted incomplete two-sample $U$-statistics is highly non-trivial: the extension of the result of Kim et al., 2022 (Minimax optimality of permutation tests, Theorem 6.3) to the case of permuted incomplete two-sample $U$-statistics is ongoing work.
> > >
> > > **Practice**: Theoretically, the cost of computing $B$ permuted estimates is $\mathcal{O}(B|\mathcal{D}_m| |\mathcal{D}_n|)$ which would be the same as if we could use a wild bootstrap. However, in practice the computational time will be much higher because for each permuted estimate we need to evaluate the kernel matrix at new permuted pairs, while for the wild bootstrap we do not need to compute any extra kernel values: this changes the computation times drastically. In order to avoid this, we would need to restrict ourselves to permutations for which we have already computed kernel values using the fact that $h_k^{MMD}(X_i, Y_s; Y_r,X_j) = - h_k^{MMD}(X_i, X_j; Y_r, Y_s)$. It remains as future work to study conditions under which the set of such permutations is larger than the set consisting of the identity only, and is also large enough to construct accurate quantiles.

---

### Official Review · Reviewer_eJnC · 2022-07-12

**Rating:** 7
**Confidence:** 4
**Soundness:** 3 good
**Presentation:** 3 good
**Contribution:** 3 good

**Summary:**

POST REBUTTAL:

Score increased to 7.

------ -------
In this work, the authors propose faster than quadratic tests for the two-sample, independence, and goodness-of-fit problems, using the Maximum Mean Discrepancy (MMD), Hilbert Schmidt Independence Criterion (HSIC), and
Kernel Stein Discrepancy (KSD), respectively. They are based on incomplete U statistics that can interpolate between linear time, and quadratic time costs (the latter cost is incurred by typical tests which are complete U-statistics).


**Questions:**


1. The work in Huggins and Mackey (e.g., the L1 IMQ and Cauchy RFF random feature Stein discrepancies) were all linear and not quadratic time as the authors mention in l 53. Given the focus on linear time tests in this work, I  believe that these tests should be treated as a useful baseline for goodness-of-fit comparison experiments. In particular, Huggins and Mackey's experiments showed that their tests typically outperformed the FSSD test (which is one of the key linear time baselines in the current work).

2. In the separation rate, the dependence on alpha is logarithmic but that on beta is polynomial (1/x)--is the latter unavoidable? I can see from the proof that it is because of the nature of concentration inequalities used in the two contexts (namely Rademacher chaos concentration, and Markov's inequality)--but are the arguments known to be tight? Does there exist a setting where such dependence is necessarily needed?

3. Is the tradeoff in Theorem 1 tight? I can see it's tight when L = N^2 but is it tight for smaller values of L? Some discussion on this would be very useful.


4. Do you not need any requirements on p for Proposition 1? And does only the difference p-q need to lie in the Sobolev ball for Thm 1(ii)?


5. Does the choice of design D not matter? Does it have to be an iid subsample? Can it be adaptive? (My guess is all the arguments go through relying on iidness of data points in D).

Minor comment:
- It would be easier to process the results if the assumptions on densities are stated in an assumption environment, and then referenced in the theorem results.

**Limitations:**

See questions.

**Strengths And Weaknesses:**

The authors provide a tradeoff between the computational cost used and the power achieved in Theorem 1---while achieving the minimax rates over Sobolev balls when using the quadratic runtime variant. The authors then use this result to also achieve appropriate power results for kernel selection (up to logarithmic inflation in the number of kernels). Notably, this result is adaptive and does not require the knowledge of the smoothness parameter of the difference between the null and the alternative density.

The authors also provide several experiments which demonstrate the advantages of the proposed methods.

The writing of the paper is pretty clear and worth appreciating!

---

> ### Author Response · Authors · 2022-08-02
> **Reply to Reviewer eJnC**
>
> We thank reviewer eJnC for his/her comments and questions.
>
> **Q1:** We thank the reviewer for catching the typo that the tests of Huggins and Mackey, 2018 are linear and not quadratic time. We have included an experiment in Appendix H.1 which compares KSDAggInc against the L1 IMQ and Cauchy RFF test of Huggins and Mackey.
>
> **Q2:** As the reviewer correctly pointed out, the polynomial factor in $\beta$ parameter comes from Markov/Chebyshev’s inequality. We believe that this polynomial dependence can be improved using more advanced concentration inequalities. Unfortunately, we do not have the right tool for proving this result at this moment and thus leave this interesting direction for future work. Nevertheless, the current sufficient conditions are sharp enough to prove optimality results in certain regimes.
>
> **Q3:** The minimax rate is the best rate that any test (of any time complexity) can achieve. As shown in Theorem 1, we can prove that the quadratic time MMD and HSIC tests achieve the minimax rate, and thus they are minimax rate optimal. To the best of our knowledge, it is unknown whether computationally efficient tests (faster than quadratic time) can achieve this rate, and `minimax rates for a given computational budget', say $L$ as a function of $N$, have not been explored in the literature. Theorem 1(ii) demonstrates a trade-off between the computational budget and the separation rate focusing on incomplete U-statistics but our result doesn’t tell us whether this trade-off is (universally) tight. We think this is one of the limitations of our work and hope that a follow-up study can make progress on this topic.
>
> **Q4:** In Proposition 1, for the two-sample and independence problems, there are no assumptions required on the densities. The proof first shows exchangeability of the wild bootstrap and original samples, and then relies on Lemma 1 of Romano and Wolf, 2005. For the goodness-of-fit setting, we need $\mathbb{E}_q[h_p^{KSD}(Z,Z)] < \infty$ and $\mathbb{E}_q\left[\left\Vert \nabla \log \frac{p(Z)}{q(Z)}\right\Vert_2^2\right] < \infty$ in order to satisfy the conditions of Theorem 2.2 of Chwialkowski et al., 2016. Those conditions are presented in the introduction, and we will repeat them in the statement of Proposition 1 in the final version.
>
> For Theorem 1(ii), we assume only that the difference $p-q$ lies in the Sobolev ball. Intuitively, we can view $q$ as a perturbed version of $p$ and we require that the perturbation is smooth (i.e. lies the a Sobolev ball).
>
> **Q5:** All results presented in the paper hold for both fixed design and design with elements sampled uniformly without replacement (the upper bound in Lemma 1 also holds for design with elements sampled uniformly with replacement). The results we prove hold for all choices of such design, however, this does not mean that the choice of design does not matter in practice. While our upper bound on the variance of the incomplete $U$-statistic holds for all choices, the variance depends on the choice of design. Certain choices of design lead to minimum variance of the incomplete $U$-statistic (see Lee, 1990, $U$ -statistics: Theory and Practice, Section 4.3.2). We are unsure how the design could be chosen adaptively, but we stress the design (or design strategy with randomness) is chosen independently of the data.
>
> We thank the reviewer for suggesting combining all assumptions in one environment to improve clarity - we will do so in the final version.

---

> > ### Comment · Reviewer_eJnC · 2022-08-06
> > **Second response**
> >
> > Thank you for your well-organized response, and for doing the additional experiments.
> >
> > I think the readers will benefit from the limitations / future directions as discussed in Q2, and Q3, and the mathematical clarifications for Q4 and Q5.

---

> > > ### Comment · Reviewer_eJnC · 2022-08-06
> > > **Second response continued**
> > >
> > > A few comments (that I believe need some clarification) regarding your new experiments:
> > >
> > > 1. Given that Cauchy RFF is the best baseline, and often performs better than your method, it would make sense to add it to your main results, and not in the appendix.
> > >
> > > 2.  How many features were used for Cauchy RFF as there will be a trade-off between running time and power? And are the runtimes reported based on the same code for all methods? (That is, oftentimes slower runtimes for a test/method might be due to worse implementation, rather than a feature intrinsic to the method/test itself).

---

> > > > ### Author Response · Authors · 2022-08-09
> > > > **Response to Reviewer eJnC**
> > > >
> > > > We thank reviewer eJnC for the further questions and for pointing out that answers provided will benefit the main paper. We will make sure to include them accordingly in the final version.
> > > >
> > > > **Q1**: We have included the additional experiments in an appendix for the rebuttal. We should have clarified that for the final version Figure 2 in Appendix H.1 will replace Figure 1 in the experiments section (Section 8).
> > > >
> > > > **Q2**: As in the experiments section of Huggins and Mackey (and as for FSSD), 10 features have been used for Cauchy RFF and L1 IMQ. We have originally used the implementation provided by Huggins and Mackey with the parameters they use in their experiments. We have noticed that they draw 5000 samples from the unnormalized density for covariance matrix estimation to simulate the null hypothesis (code: RFDH0SimCovDrawV(n_draw=5000)). This procedure causes the long runtimes observed, it is much more expensive than simulating the null using a wild bootstrap as KSDAggInc does.
> > > >
> > > > We have tried different values for n_draw. Using n_draw=500 has almost no effect on test power (minor decrease) and reduces the runtimes from 16 seconds for n_draw=5000 to 2 seconds for n_draw=500. We tried smaller values than 500 for n_draw but this resulted in a significant decrease in test power, we have also verified that the test still has well-calibrated level. We have added a row to Figure 2 in Appendix H.1 running Cauchy RFF and L1 IMQ with n_draw=500. We have also added a new figure (Figure 3) to Appendix H.1 where we compare KSDAggInc $R=200$ with Cauchy RFF and L1 IMQ with n_draw=500,5000.
> > > >
> > > > Overall, KSDAggInc and Cauchy RFF (with n_draw = 500 or n_draw = 5000) obtain the same performance in terms of test power. While KSDAggInc runs faster in the experiments presented, even with the much lower n_draw, it seems that the KSDAggInc runtimes increase more steeply with the sample size than the Cauchy RFF / L1 IMQ runtimes. Note that the power of KSDAggInc could be improved by increasing $R$ (i.e. increasing $c$ in the design size $cN$) but it is of course upper bounded by the power of KSDAggCom. The code has been updated on the original anonymized repository (link line 74).
> > > >
> > > > In Figure 2 (and Figure 1 and 3), the time plots (4th column) correspond to the experiments run by varying the sample size in the first column. As detailed in Appendix B lines 574/575: 'In Figure 1(i,l) [same for Figure 2(i,l)], we consider dimensions $d_x = 50$ and $d_h = 40$ with noise standard deviation $\sigma = 0.02$ and we vary the sample size $N \in \{200, 400, 600, 800, 1000\}$.
> > > >
> > > > Runtimes reported for KSDAggInc are based on our implementation (anonymized repository line 74), runtimes reported for FSSD are based on the implementation provided in their paper (kernel-gof repository by Wittawat Jitkrittum), runtimes reported for L1IMQ and Cauchy RFF are based on the implementation provided in their paper (random-feature-stein-discrepancies repository by Jonathan Huggins). We will stress that the plots show runtimes of the tests obtained in practice when using the implementations provided by the respective authors of the tests, but that the observed speed difference might be due to implementation (the tests are not theoretically shown to be faster/slower).
> > > >
> > > > We hope this analysis addresses the questions raised, and if so, that reviewer eJnC will consider increasing their score.

---

### Official Review · Reviewer_STxK · 2022-07-12

**Rating:** 7
**Confidence:** 3
**Soundness:** 3 good
**Presentation:** 3 good
**Contribution:** 3 good

**Summary:**

This paper studies a family of nonparametric two-sample, independence, and goodness-of-fit tests based on incomplete kernel-based U-statistics, for which it proves validity (Proposition 1) as well as guarantees on power, assuming the true densities lie in a Sobolev space and are sufficiently well separated in $L_2$ distance. The power guarantees are initially proven for a statistic depending on the true smoothness of the Sobolev space (Theorem 1), but Theorem 2 extends this to an estimate that is adaptive to unknown smoothness. Theorem 3 shows that, compared to existing results for independence testing, a tighter bound, with better dependence on the type-1 error probability, can be obtained. Finally, some experiments demonstrate how the performance of the proposed estimators varies with hyperparameters and how this compares with some other linear-time nonparametric tests.

**Questions:**

1. Lemma 1: The variance bound for random design includes a $\frac{1}{|\cal{D}_r|} + \frac{1}{N^2}$ term. Since $|\cal{D}_r| \leq N^2$, isn’t the second term redundant (i.e., can’t it be absorbed into the constant $C$)?
3. Line 144-146, “The motivation for defining the estimators… of order 2 (rather than of higher order) derives from the reasoning of Kim et al. (2022, Section 6)...”: I didn’t quite understand this sentence. I skimmed Section 6 of Kim et al. (2022), and, while they do indeed study U-statistics of order 2, the motivation for order 2 (rather than of higher order) wasn’t obvious to me. Could the authors clarify?
2. I found the motivation for Theorem 3 (lines 241-256) a bit hard to understand. Am I understanding correctly that the improved (logarithmic rather than polynomial) dependence on $1/\alpha$ has been previously shown for two-sample testing but not for independence testing. Later on (Lines 262-263), the paper says “As discussed by Kim et al. (2022, Section 8.3), their proposed sample-splitting method can also be used to obtain the correct dependency on $\alpha$.” So what exactly is the new contribution of Theorem 3?
4. Figure 1: The first row of plots includes a green curve that isn’t included in the legend. What is this? Also, the paper discusses results for some methods (e.g., OST PSI) for which I didn’t see any results in Figure 1. Where are these results reported?
5. Could the authors elaborate on advantages of the proposed tests over previous tests that have been shown to be minimax optimal (e.g., the Gaussian-kernel-based tests of Li and Yuan (2019))?

**Limitations:**

The paper would definitely benefit from further discussion of the limitations of its present results and suggestions for future work. However, given space limitations, I don't think further discussion of this is strictly necessary for acceptance.

**Strengths And Weaknesses:**

Strengths: This paper provides much more compelling theoretical results (minimax optimality, especially for an adaptive estimator (Theorem 2)) than most related work on two-sample testing, which usually only shows consistency. It’s also nice that a unified but rigorous discussion is given for the closely related problems of two-sample, independence, and goodness-of-fit testing.

Weaknesses: The paper has a lot of notation that is very similar or overloaded, but not explained or disambiguated near where it is used. For example, $L$ is defined just before Line 177 as the design size of the incomplete U-statistic, but is also used as a kernel (in Eq. (12)) and for $L^p$ spaces (in Eq. (17)). This made it a bit hard for me to follow the paper’s notation. I think it would help if the paper was a bit more explicit (even redundant) with explaining its notation near where it is used (e.g., reiterating “where $L$ is the design size” after Theorem 1). The paper also is not particularly clear about its distinctions from prior work (see questions below, although I was able to piece this together from various parts of the paper and by reading some of the references).

---

> ### Author Response · Authors · 2022-08-02
> **Reply to Reviewer STxK**
>
> **Q1:** The term $\frac{1}{N^2}$ can indeed be upper bounded by $\frac{1}{|\mathcal{D}_r|}$ as done in the proof of Theorem 1, which corresponds to absorbing $\frac{1}{N^2}$ in the constant. Thank you for pointing this out! This will simplify the statement of Lemma 1.
>
> **Q2:** For the independence problem, we have pairs of samples $(X_i,Y_i)_{i=1}^N$. The classical HSIC permuted $U$-statistic
> In Equation (26) of Kim et al. (2022, Section 6), the HSIC permuted $U$-statistic is defined. Fixing the permutation to the identity, this gives the HSIC $U$-statistic
> $$
> U_1
> =
> \frac{1}{\mid\textbf{i}_2^N\mid}
> \frac{1}{\mid\textbf{i}_2^N\mid}
> \sum^{(i,j)\in \textbf{i}_2^N}
> \sum^{(r,s)\in \textbf{i}_2^N}
> h^{HSIC}(Z^i,Z^j;Z^r,Z^s)
> $$
> Now, one way to construct an incomplete HSIC $U$-statistic would be to replace those two complete sums with two incomplete sums, but we do not want to do this in order to keep a unified framework across the three testing frameworks.
> So, instead we pair the variables with index $a$ and $a+\lfloor N/2\rfloor$ for $a=1,\dots,\lfloor N/2\rfloor$ to obtain an estimator
> $$
> U_2
> =
> \frac{1}{\mid\textbf{i}_2^{\lfloor N/2\rfloor}\mid}
> \sum^{(a,b)\in \textbf{i}_2^N}
> h^{HSIC}(Z^a,Z^b;Z^{a+\lfloor N/2\rfloor},Z^{b+\lfloor N/2\rfloor})
> =
> \frac{1}{\mid\textbf{i}_2^{\lfloor N/2\rfloor}\mid}
> \sum^{(a,b)\in \textbf{i}_2^N}
> h^{HSIC}(Z^{a+\lfloor N/2\rfloor},Z^{b+\lfloor N/2\rfloor};Z^a,Z^b)
> $$
> This corresponds to the discussion following Equation (26) of Kim et al., 2022 with the $\lfloor N/2\rfloor$-tuple $L\coloneqq \{1,\dots,\lfloor N/2\rfloor\}$ when the permutation is the identity, where $L$ is the notation used in Kim et al., 2022. In Equation (27) they then show that the expectation of $U_2$ with respect to the uniform choice of $L$ is $U_1$.
> This motivated our choice of HSIC estimate in Equation (8) of our paper.
>
> **Q3:** Yes, the improved (logarithmic rather than polynomial) dependence on $1/\alpha$ has been shown for two-sample testing but not for independence testing based on the permutation procedure “without sample splitting”. As we briefly mentioned, the logarithmic dependence on $1/\alpha$ is possible by converting the independence testing problem into the two-sample problem via sample-splitting. This was the main idea proposed in Section 8 of Kim et al., (2022). While this indirect approach leads to a logarithmic factor in \alpha, the practical power would be suboptimal due to an inefficient use of the data from sample splitting. Our result is based on the standard U-statistic for independence testing calibrated by the usual permutation approach, which does not depend on sample splitting. In particular, our result shows that the usual permutation-based HSIC test has the same logarithmic dependence in $\alpha$ with the less practical test in Kim et al., (2022). In the final version, we will revise Section 7 to make our contributions clearer.
>
> **Q4:** The first row of plots (Figure 1) includes a green curve which is the third one in the legend (between SCF and MMDAggInc R=1) and corresponds to OST PSI.
>
> **Q5:** Li and Yuan (On the optimality of gaussian kernel based nonparametric tests against smooth alternatives, 2019) also consider the three testing problems and show minimax optimality/adaptivity over Sobolev balls. Their tests run in quadratic time and control the probability of type I error only asymptotically, while our proposed tests have well-calibrated non-asymptotic levels over a broader class of null distributions and are computationally efficient. Their theoretical guarantees hold only for the Gaussian kernel and with the smoothness restriction that $s>d/4$ while ours hold for a wide range of kernels (see Equation (12)) and for any $s>0$ (see Theorem 2). Note that, they tackle the goodness-of-fit problem in a different way. They do not use the KSD and instead use a one-sample MMD with some expectations of the Gaussian kernel under the model. For a generic model density, one cannot compute such an expectation and hence cannot use their proposed test, while it is possible to use the KSD (which makes the kernel expectation under the model vanish).
>
> **Future work:** Potential directions for future work include studying the regime with $L \lesssim N$, which corresponds to 'faster than linear' tests. For this sub-linear case, our results do not give a definite answer to the question as to whether the upper bound converges to zero. Future work would focus on either deriving tighter bounds which prove convergence to zero in this regime, or proving that the uniform separation rate diverges in this setting. Another interesting direction for future work is to see whether it is possible to achieve minimax rate optimality in sub-quadratic time complexity. Also, it would be interesting to see if the polynomial factor of $\beta$ in our condition can be improved using a sharper concentration bound. Due to page limit, we will discuss future directions and limitations of our proposals in the appendix.

---

> > ### Comment · Reviewer_STxK · 2022-08-08
> > **Thanks to the authors for their response**
> >
> > The author response addressed my questions, thank you. I will retain my scores, and ask that the authors clarify the above points, especially Q3, in the final version of the paper (except Q4, where it seems I had somehow missed the figure legend).

---

> > > ### Author Response · Authors · 2022-08-09
> > > **Response to Reviewer STxK**
> > >
> > > We thank the reviewer once again for their feedback.
> > >
> > > We will include the discussed points in the final version of the paper.

---

### Author Response · Authors · 2022-08-02
**Comment to all reviewers**

We warmly thank all reviewers for their careful reading of our paper and their invaluable insights. We particularly thank reviewer **STxK** for emphasizing that our work provides 'unified but rigorous discussion' with 'much more compelling theoretical guarantees [...] than most related work [...] which usually only shows consistency'; reviewer **eJnC** for pointing out that 'the writing of the paper is pretty clear and worth appreciating'; and reviewer **aMao** for noting that 'the provided code is clean and it is easy to reproduce the experiments'.

We individually answer the questions of the reviewers below. We have provided additional experiments in Appendix H. We hope these address the concerns expressed by the reviewers, and if so, that they would kindly consider upgrading their evaluation. We provide here a brief discussion of the additional experiments (the code for reproducibility is provided on the original anonymised github repo).

**Large collection of bandwidths** (Appendix H.1): We have increased the number of bandwidths our tests aggregate over. For MMDAggInc and KSDAggInc, we aggregate over 21 bandwidths which are $\{2^i \lambda_{med} : i = -10,\dots,10\}$. For KSDAggInc, we aggregate over 25 bandwidths $\{(2^i \lambda_{med}, 2^j \lambda_{med}) : i,j = -2,\dots,2\}$. Firstly, our new simulation results show that the proposed tests retain high power even when a large collection of bandwidths is used. Secondly, we believe that this revised approach mitigates the concern around the collection of bandwidths: loosely speaking, for the Gaussian kernel, we are essentially aggregating over kernel matrices which interpolate between the identity matrix (very small bandwidth) and the matrix of ones (very large bandwidth).

**Compare against Huggins and Mackey, 2018** (Appendix H.1): We compare KSDAggInc against the L1_IMQ test and Cauchy RFF of Huggins of Mackey (Random Feature Stein Discrepancies, 2018). L1IMQ performs similarly to the FSSD test in our RBM experiment, which is coherent with the results presented in Figure 4a of Huggins and Mackey, 2018. Cauchy RFF performs only very slightly better than our proposed test KSDAggInc $R=200$ but takes much longer to run (16 seconds against less than a second).

**Benefits of aggregation** (Appendix H.2): We illustrate the benefits of the aggregation procedure by starting from a 'collection' consisting of only the median bandwidth and increasing the collection by adding more bandwidths. In all three settings, we observe that the power for the test with the median bandwidth only is low. As we increase the number of bandwidths, the power first increases as the test has access to 'better bandwidths'. For MMDAggInc and KSDAggInc: once the optimal bandwidth is included in the bandwidth, the power decreases slightly and reaches a plateau. HSICAggInc is more challenging, since there are kernels for both X and Y, hence the total number of bandwidth combinations grows rapidly (9 bandwidths for each of X and Y = 81 total combinations). For this case, we do experience a decay in test power once many bandwidths are considered, due to the large number of such combinations.

**Experiments on MNIST dataset** (Appendix H.3): We demonstrate in experiments using the real-world MNIST dataset (dimension 784) that our proposed tests also obtain higher power than the tests we compare against.

---

### Meta-Review · Area_Chair_dguL · 2022-08-26

**Recommendation:** Accept
**Confidence:** Certain

**Metareview:**

The paper discusses fast computation methods for kernel-based statistical tests: MMD, HSIC, and KSD.  The paper uses incomplete U statistics in constructing the methods, shows decent theoretical results including the rate analysis, and confirms favorable numerical results.  The paper has significant theoretical contributions to the topic, and also demonstrates the practical usefulness of the methods.  After the revision, all the reviewers agree to accept this paper to NeurIPS.


**Award:**

No

---

### Decision · Program_Chairs · 2022-09-14

Accept